# The $s$-value: evaluating stability with respect to distributional shifts

**Suyash Gupta**
Department of Statistics
Stanford University
Stanford, CA 94305
suyash028@gmail.com

**Dominik Rothenhäusler**
Department of Statistics
Stanford University
Stanford, CA 94305
rdominik@stanford.edu

## Abstract

Common statistical measures of uncertainty such as $p$-values and confidence intervals quantify the uncertainty due to sampling, that is, the uncertainty due to not observing the full population. However, sampling is not the only source of uncertainty. In practice, distributions change between locations and across time. This makes it difficult to gather knowledge that transfers across data sets. We propose a measure of instability that quantifies the distributional instability of a statistical parameter with respect to Kullback-Leibler divergence, that is, the sensitivity of the parameter under general distributional perturbations within a Kullback-Leibler divergence ball. In addition, we quantify the instability of parameters with respect to directional or variable-specific shifts. Measuring instability with respect to directional shifts can be used to detect under which kind of distribution shifts a statistical conclusion might be reversed. We discuss how such knowledge can inform data collection for transfer learning of statistical parameters under shifted distributions. We evaluate the performance of the proposed measure on real data and show that it can elucidate the distributional instability of a parameter with respect to certain shifts and can be used to improve estimation accuracy under shifted distributions.

## 1 Introduction

Test data sets collected in different locations or at different time points often are drawn from different distributions, due to changing circumstances, changes in unmeasured confounders, time shifts in distribution, or distributional shifts in covariates [44, 19, 16, 20]. This makes it difficult to gather knowledge that transfers across data sets. Statistical estimands such as a regression coefficient or the average treatment effect (ATE) may vary as the underlying distribution changes and hence, statistical findings (such as that the treatment effect is positive) may not replicate across data sets [4, 22].

In causal inference, the rapidly growing field of sensitivity analysis [12, 40, 16, 53, 11] quantifies the stability of an estimate with respect to unobserved confounding. Roughly speaking, this line of work sees stability analysis as part of uncertainty quantification. Inspired by this line of work, we aim to bring a similar type of stability analysis to a wider range of statistical procedures.

In this paper, we propose a measure of instability, called the $s$-value, to investigate the stability of a given statistical parameter with respect to a shift in the underlying distribution (Figure 1). The $s$-value quantifies the minimum shift in distribution required to tilt the parameter to a given value, using Kullback-Leibler divergence. We also investigate the stability of parameters with respect to directional or variable-specific shifts. The proposed measure can be used as an exploratory tool to identify the kind of distribution shift that could reverse a statistical conclusion. We further discuss

37th Conference on Neural Information Processing Systems (NeurIPS 2023).

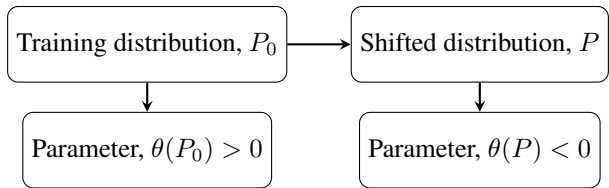

Figure 1: Distribution shift can change the parameter of interest.

how $s$-values can be used to obtain improved estimates of statistical parameters under a shifted distribution with limited information about the new distribution.

### 1.1 Our contribution

We propose a measure of instability that quantifies the sensitivity of a one-dimensional statistical parameter to changes in the underlying probability distribution. We focus on shifts in distributions that are absolutely continuous with respect to the training distribution. Let $P_0 \in \mathcal{P}$ be the training distribution on the measure space $(\mathcal{Z}, \mathcal{A})$, where $\mathcal{P}$ is the set of probability measures, $Z$ is a random element of $\mathcal{Z}$, and $\theta : \mathcal{P} \mapsto \mathbb{R}$ is the one-dimensional statistical parameter of interest. We are interested in the minimum amount of shift in distribution that changes the sign of the parameter. To this end, we define the stability value ($s$-value) for $\theta$ as

$$s(\theta, P_0) = \sup_{P \in \mathcal{P}} \exp -D_{KL}(P \parallel P_0) \quad \text{s.t.} \quad \theta(P) = 0, \tag{1}$$

where $D_{KL}$ is the Kullback-Leibler divergence between $P$ and $P_0$ given by

$$D_{KL}(P \parallel P_0) = \int \log \left( \frac{dP}{dP_0} \right) dP.$$

We provide some more discussion on the thought process that led to the proposed definition in Appendix, Section A. Note that the $s$-value lies in $[0, 1]$, with values close to 1 indicating that a small shift in distribution may alter the findings, and hence, the finding is not distributionally stable. $S$-values close to 0 indicate that the sign of the parameter is stable under distributional changes. In Section 3, we discuss estimation of $s$-values for parameters that are linear in the distribution $P$. The proposed procedure can be generalized to parameters defined via risk minimization, including parameters in generalized linear models (see Appendix, Section D).

Considering overall distributional shift does not give information about what kind of distribution shifts the parameter is sensitive to. Hence, we also quantify the instability of parameters with respect to shifts in the distribution of certain exogenous or endogenous variables $E$, assuming that the conditional distribution of the remaining variables given $E$ is constant. Let $E$ be a random variable taking values in the space $\mathcal{E}$. We define the directional or variable-specific $s$-value as

$$s_E(\theta, P_0) = \sup_{P \in \mathcal{P}: P(\cdot|E=e) = P_0(\cdot|E=e) \text{ for all } e \in \mathcal{E}} \exp -D_{KL}(P \parallel P_0) \quad \text{s.t.} \quad \theta(P) = 0. \tag{2}$$

where $\mathcal{P}$ denotes the set of probability distributions over joint random variable $(Z, E)$. If a practitioner discovers that a parameter is sensitive with respect to changes in the distribution of a certain variable $E$, this knowledge can be used to update the parameter estimate. We discuss how our method can be used to prioritize data collection about the new distribution in Section 4 and use the same to re-estimate the parameter under shifted distribution. The proposed procedure shows promise for the task of prioritizing data collection from the new distribution in the experiments.

## 2 Related work

Quantifying the uncertainty of statistical estimators is a crucial objective in statistics, typically accomplished using classical statistical measures such as $p$-values and confidence intervals to quantify sampling uncertainty. However, these methods typically rely on strong, potentially unjustified

assumptions about fixed underlying distributions, which may lead to false discoveries. To improve reliability and reproducibility in statistical estimation, Yu and Kumbier [52] propose the predictability, computability, and stability (PCS) framework. While they investigate the stability of data results under data and method perturbations, we focus specifically on evaluating the stability of statistical parameters under distributional shifts.

Model misspecification can result in distributional instability. Buja et al. [6] highlight fundamental issues with model misspecification or non-linearity in linear models. They propose reinterpreting population slopes as statistical functionals of data generating distributions and develop diagnostic tests for detecting model deviations. We introduce measures to illustrate coefficient instability under various distributional shifts for parametric and semi-parametric estimators.

[34] introduce a novel framework to analyze the stability of decision policies and prediction models under distribution shifts. Central to their approach is the notion of stability, characterized as the minimal alteration in the underlying environment required to push a system's performance beyond a specified threshold. In contrast, we focus on understanding stability of parameters with respect to shift in distribution.

In machine learning, there is much work on computing which data or features contribute to a prediction, e.g. using Shapley values [33, 21]. In contrast, we are interested in how distributional changes in features (or covariates) lead to parameter changes.

Sensitivity analyses in the causal inference literature aim to investigate the stability of causal estimates with respect to unmeasured confounding [12, 40, 16, 53, 11]. Our proposal can be seen as a version of sensitivity analysis for general estimands where we evaluate both the stability of an estimand with respect to the overall shift in distribution and the stability with respect to directional distribution shifts.

Classical robust statistics [24] addresses robustness against contaminations and outliers using measures like leverage scores and influence functions to construct estimators that are not unduly influenced by such outliers. Influence functions play an important role in this work, since it corresponds to the functional derivative of a parameter with respect to the distribution. Different from classical robust statistics, in our case the perturbation is not a contamination but corresponds to an actual change in the underlying population. Rather than robustifying estimators, our aim is to equip practitioners with tools to detect sources of instability and facilitate the transfer of estimators across different settings.

There has been a resurgence of research addressing the challenges posed by distributional shifts. This research has mostly focused on building distributionally robust estimators where more weight is given to the outliers by considering worst-case distributional shifts in a neighborhood of the training distribution [19, 45, 20, 43, 25, 7, 9, 8, 47]. In contrast, we quantify the stability of potentially non-linear statistical parameters under both overall and variable-specific distributional shifts.

There is exciting empirical work by Devaux and Egami [15] who build a library of reference stability values based on survey data sets. They recommend thresholds for s-values based on empirical investigations of how data sets change between settings. This work allows for the contextualization of distributional stability values.

Closely related to our work are empirical likelihoods [35]. In the empirical likelihood framework, small overall distributional tilts are used to construct $p$-values and confidence intervals for a given parameter. In our work, we use distributional shifts of various strengths to evaluate the (directional) stability of an estimand with respect to distribution shifts.

## 3   S-value of the mean

In this section, we discuss estimation of the $s$-value of the mean of a one-dimensional real-valued random variable followed by some examples. Estimation of $s$-values for more general settings, including parameters defined via risk minimization, is discussed in Appendix, Section D. We first focus on the special case of mean estimation as it allows us to develop procedures that will be helpful in more general settings.

## 3.1 Estimation of the $s$-value

Consider a one-dimensional real-valued random variable $Z \sim P_0$, where $P_0 \in \mathcal{P}$. We recall from (1) that the $s$-value for the mean $(\mu(P_0) = \mathbb{E}_{P_0}[Z])$ is defined as

$$s(\mu, P_0) = \sup_P \exp\{-D_{KL}(P||P_0)\} \quad \text{s.t.} \quad \mathbb{E}_P[Z] = 0. \tag{3}$$

In words, we are interested in finding the distribution closest to our training distribution $P_0$ under which the mean of the random variable is $0$. At first sight, $s$-values might seem difficult to estimate since the supremum in equation 3 is taken over the infinite-dimensional space of probability distributions $\mathcal{P}$. However, it turns out that the $s$-value of the mean can be obtained by solving a one-dimensional convex optimization problem.

**Theorem 1** (Theorem 5.2, Donsker and Varadhan [17]). *Let $Z \sim P_0$ be a real-valued random variable with mean $\mu(P_0) = \mathbb{E}_{P_0}[Z]$ and finite moment generating function on $\mathbb{R}$. Then, we have*

$$s(\mu, P_0) = \inf_\lambda \mathbb{E}_{P_0}[e^{\lambda Z}]. \tag{4}$$

*Further, if the infimum in (4) is attained at some $\lambda^* \in \mathbb{R}$ then the infimum in (3) is attained at some probability distribution $Q$ given by*

$$dQ(z) = \frac{e^{\lambda^* z}}{\mathbb{E}_{P_0}[e^{\lambda^* Z}]} dP_0(z) \text{ for all } z \in \mathbb{R}.$$

We note that $M_Z(\lambda) = \mathbb{E}[e^{\lambda Z}]$ is the moment generating function of $Z$. Since, $M_Z(0) = 1$, we have $s(\mu, P_0) \in [0, 1]$. In practice, we only have access to finitely many realizations of the data generating distribution. Let $P_n$ be the empirical distribution of $Z_i \overset{\text{i.i.d.}}{\sim} P_0$ for $i \in [n]$, we obtain an estimator of the $s$-value via the plugin estimator

$$\hat{s}(\mu, P_n) = \inf_\lambda \mathbb{E}_{P_n}[e^{\lambda Z}] = \inf_\lambda \frac{1}{n} \sum_{i=1}^n e^{\lambda Z_i}. \tag{5}$$

Using classical results for $M$-estimators (see Chapter 5 of van der Vaart [48]), we show that $\hat{s}(\mu, P_n)$ is consistent and asymptotically normal in Appendix C.1.

**Directional $s$-values.** The previous form of distributional stability might be very conservative. In practice, we do not expect all aspects of distribution to change from setting to setting. To allow for a more fine-grained evaluation of stability, we also consider directional shifts, which only change certain aspects of the distribution. In the following, we will make this more precise.

Let $P_0$ be the joint distribution of the multivariate random variable $(Z, E)$ where $Z$ takes values in $\mathcal{Z} \subseteq \mathbb{R}$ and $E$ takes values in $\mathcal{E} \subseteq \mathbb{R}^p$ for some positive integer $p$. $E$ may be an exogenous or endogenous variable. We consider a directional shift, i.e. a situation where the marginal distribution of $E$ may change while keeping the conditional distribution of $Z$ given $E$ constant. To be more precise, we seek to estimate

$$s_E(\theta, P_0) = \sup_{P \in \mathcal{P}: P(\cdot | E=e) = P_0(\cdot | E=e) \text{ for all } e \in \mathcal{E}} \exp\{-D_{KL}(P||P_0)\} \quad \text{s.t.} \quad \theta(P) = 0. \tag{6}$$

We next show that $s_E$ is a solution to a one-dimensional convex optimization problem. The proof of the following result can be found in Appendix I.2.

**Theorem 2.** *Let $P_0$ be the joint distribution function of the random variable $(Z, E)$ taking values in $\mathcal{Z} \times \mathcal{E}$ with $\mu = \mathbb{E}_{P_0}[Z]$ and finite moment generating function on $\mathbb{R}$. Then,*

$$s_E(\mu, P_0) = \inf_\lambda \mathbb{E}_{P_0}[e^{\lambda \mathbb{E}_{P_0}[Z|E]}]. \tag{7}$$

*Further, if the infimum in (7) is attained at some $\lambda^* \in \mathbb{R}$ then the infimum in (2) is attained at some probability distribution $Q$ given by*

$$dQ(z, e) = \frac{e^{\lambda^* \mathbb{E}_{P_0}[Z|E=e]}}{\mathbb{E}_{P_0}[e^{\lambda^* \mathbb{E}_{P_0}[Z|E]}]} dP_0(z, e) \text{ for all } (z, e) \in \mathcal{Z} \times \mathcal{E}.$$

This result allows us to estimate the directional $s$-value. Let $\hat{f}_n(E)$ be an estimator of $\mathbb{E}[Z \mid E]$. Then, we can define a plug-in estimator by setting

$$\hat{s}_E(\mu, P_n) = \inf_{\lambda} \frac{1}{n} \sum_{i=1}^{n} e^{\lambda \hat{f}_n(E_i)}. \tag{8}$$

We prove the consistency of $\hat{s}_E(\mu, P_n)$ in Appendix, Section C.1. In the Appendix, we also discuss how to form a de-biased estimator of the directional $s$-value that is asymptotically normal.

### 3.2 Examples

**Example 1** (Distribution with positive support). *If $Z$ is a random variable that has positive support with probability 1, then $s(\mu, P_0) = 0$, which reflects the fact that for any distribution shift within the KL-divergence ball, we will always have a positive mean.*

**Example 2** (Gaussian distribution). *If $Z \sim N(\mu, \sigma^2)$, then $s(\mu, P_0) = e^{-\frac{\mu^2}{2\sigma^2}}$.*

Thus, in the Gaussian case the stability measure $s$ is a monotonous transformation of the signal-to-noise ratio. High signal-to-noise ratio yields lower values of $s$ indicating stronger distributional stability.

Let us now develop some intuition for directional shifts. First, we derive conditions under which the directional stability is zero, that is, conditions under which $s_E(\mu, P_0) = 0$.

**Example 3** (Directional stability). *Let $\mathbb{E}_{P_0}[Z|E] > 0$. Then,*

$$s_E(\mu, P_0) = \inf_{\lambda} \mathbb{E}_{P_0}[e^{(\lambda \mathbb{E}_{P_0}[Z|E])}] = \lim_{\lambda \to -\infty} \mathbb{E}_{P_0}[e^{(\lambda \mathbb{E}_{P_0}[Z|E])}] = 0.$$

**Example 4** (Average treatment effect). *Here we consider estimating the causal effect of a treatment via the potential outcome framework [46, 42]. We have a binary treatment random variable $A \in \{0, 1\}$, potential outcomes $Y(1)$ and $Y(0)$ corresponding to the potential outcome under treatment and control respectively and some covariates $X$. Under the consistency assumption, we observe $Y(1)$ if $A = 1$ and $Y(0)$ if $A = 0$, i.e. $Y = AY(1) + (1 - A)Y(0)$. One can write the average treatment effect (ATE) as*

$$\tau = \mathbb{E}_{X \sim P_X} \mathbb{E}[Y(1) - Y(0) \mid X] = \mathbb{E}_{X \sim P_X}[\mu_{(1)}(X) - \mu_{(0)}(X)],$$

*where $\mu_{(a)}(X) = E[Y(a) \mid X]$. Hence, if we only consider shifts in marginal distribution of covariates $X$ keeping the conditional distribution of other variables given the covariates as fixed, we obtain $s_X$-values as above with $Z = \mu_{(1)}(X) - \mu_{(0)}(X)$. In practice, $\mu_{(1)}(X)$ and $\mu_{(0)}(X)$ are often unknown. We can use plug-in estimators $\hat{\mu}_{(1)}(X)$ and $\hat{\mu}_{(0)}(X)$ to form the estimator*

$$\hat{s}_X(\tau, P_0) = \inf_{\lambda} \frac{1}{n} \sum_{i=1}^{n} e^{\lambda(\hat{\mu}_{(1)}(X_i) - \hat{\mu}_{(0)}(X_i))}.$$

*Consistency of this estimator can be shown with the same technique as Lemma C.3 in Appendix C.1.*

In statistical analysis, risk minimization is often used to define various parameters, including regression coefficients and general M-estimators. However, unlike parameters that can be represented as the mean of a random variable, these parameters lack a simple representation as they are not linear in the underlying probability distribution. This makes the optimization problem involved in obtaining $s$-values non-convex. To address this issue, we present methods for obtaining $s$-values for such parameters in Section D of the Appendix.

## 4 Parameter transfer using $s$-values

In Section 3, we introduced $s$-values that measure the distributional stability of statistical parameters with respect to various shifts. In this section, we discuss how we can use $s$-values to guide further data collection. The above problem of re-estimating parameters under a shifted distribution is related to the transfer learning literature that overlaps with various fields including robust machine learning,

causal inference, and conformal inference [36, 51, 3]. Here, we discuss how $s$-values can guide transfer learning.

If a parameter is unstable with respect to a shift in marginal distribution of certain covariates, then knowledge about those covariates can be used to transfer parameters across distributions. As an example, assume that we have collected some data on a job program in New York. We now want to estimate how efficient this job program would be in Boston. We have not run this job program in Boston yet, so we do not know all covariates of the participants. However, we can find that the efficiency of the job program is likely unstable with respect to changes in the demographics of job seekers in Boston. How can we use this knowledge to estimate the efficiency of the job program in Boston, based on limited data about the population in Boston? In the following, we will discuss this problem in a formal framework. We discuss another important case. Researchers are often interested in a causal effect estimate for a new location. For some covariates such as age and education, partial data is available via surveys such as American National Election Study (ANES) or Cooperative Election Study (CES). Additional partial data can be cheaply obtained via Amazon Mechanical Turk. However, some covariates are hard to collect, since they require running a study in the new location. Our numerical results show that the proposed approach can help prioritize data collection. This may drastically reduce the cost compared to running full-scale replication studies.

Assume that we want to estimate a parameter $\theta(P_{\text{shift}})$ for $P_{\text{shift}} \neq P_0$, but we only have observations from $P_0$. In addition, we may be able to collect some information about $P_{\text{shift}}$, for example, observations of a subset of variables $X_S \in \mathbb{R}^d$. For example, one may know the age distribution of job seekers in Boston. Intuitively, we'd like to re-weight $P_0$ so that the distribution of age matches the distribution of age in Boston. However, there may be infinitely many choices of weights. These different choices of weights will correspond to different values of $\theta$. Thus, in practice, it is crucial to use a form of regularization when finding a re-weighted distribution ($P_{proj}$).

We can define $P_{proj}$ as the solution of the following optimization problem:

$$P_{proj} = \arg\min_{P'} D_{KL}(P'|P_0) \text{ such that } P'(X_S = \cdot) = P_{\text{shift}}(X_S = \cdot),$$

where $D_{KL}$ is the Kullback-Leibler divergence. The objective is similar to Hainmueller [23], where the author proposes entropy balancing to achieve covariate balance between treated and control sets for estimating the average treatment effect. This objective can be solved explicitly, leading to a covariate shift setting [36]. More specifically, a short calculation shows that if the minimum is finite, then

$$\mathrm{d}P_{proj}(z, x_S) = \mathrm{d}P_0(z|x_S)\mathrm{d}P_{\text{shift}}(x_S).$$

Since data collection can be costly, one would like to prioritize collecting data that is relevant for the transfer learning task. If $s_{X_S}(\theta - c, P) = 0$ for all $c \neq \theta(P)$, then the parameter is constant under shifts in the marginal distribution of $X_S$. On the other hand, if $s_{X_S}(\theta, P) \approx 1$, then small changes in the distribution of $X_S$ might induce a large change in the parameter $\theta(\cdot)$. These heuristics motivate the following approach:

1. Find variables $X_S$ with respect to which the parameter of interest is most sensitive to as determined by directional $s$-values. Collect observations of $X_S$ under the shifted distribution.
2. Estimate $\theta(P_{\text{proj}})$.

There are several existing approaches to deal with part 2. In particular, it is possible to estimate $\theta(P_{\text{proj}})$ for a large range of estimands $\theta(\bullet)$, with asymptotically normal and efficient estimators [38, 32, 28]. We can leverage these existing estimators in our workflow so that we have statistical guarantees for all steps in this pipeline.

We will investigate the empirical performance of this two-stage approach in Section 5.

## 5 Experiments

In this section, we consider real-world data to illustrate the effectiveness of the proposed methods in elucidating the distributional instability of various statistical procedures. In addition, we evaluate the two-stage transfer learning procedure described in Section 4.

We note that $s$-values can be used to create sensitivity plots. Under various distribution shifts, the parameter can attain a range of values. More concretely, for different choices of $E$ and an upper bound on the distribution shift $c \in \mathbb{R}$ we define upper and lower bounds for parameter values as follows:

$$\theta_{\text{upper-bound}} = \sup \theta(P) \text{ such that}$$
$$P \in \mathcal{P} : P(\cdot | E = e) = P_0(\cdot | E = e) \text{ for all } e \in \mathcal{E} \text{ and } D_{KL}(P \| P_0) \leq c$$
$$\theta_{\text{lower-bound}} = \inf \theta(P) \text{ such that} \tag{9}$$
$$P \in \mathcal{P} : P(\cdot | E = e) = P_0(\cdot | E = e) \text{ for all } e \in \mathcal{E} \text{ and } D_{KL}(P \| P_0) \leq c.$$

In experiments, we plot estimated versions of these upper and lower bounds across $c$ for different choices of the variable $E$. Note that if $\theta(P_0) > 0$, then $s(\theta, P_0)$ is the minimum $c$ for which $\theta_{\text{lower-bound}} = 0$.

## 5.1 National supported work demonstration data (NSW)

Here, we analyze the stability of the average treatment effect estimator in the presence of covariate shift using the NSW dataset [29], which consists of $n = 722$ participants randomly assigned to a treatment or control group (variable $A$) in an employment program field experiment conducted between January 1976 and July 1977. Covariates $X$ include 'age', 'education', 'black', 'hispanic', 'married', 'nodegree', and 're75', where 're75' denotes pre-intervention earnings in 1975. The outcome variable is 're78', corresponding to post-intervention earnings in 1978. We apply augmented inverse probability weighting (AIPW) using causal forests [50] to estimate the average treatment effect, resulting in an estimate of $820$ with a standard deviation of $492$. We evaluate the performance of the two-stage transfer learning procedure presented in Section 4.

**Distributional stability of average treatment effect.** We investigate the stability of the average treatment effect estimator under distributional changes. Specifically, we examine how $\mathbb{E}_P[\tau(X)]$ changes when there is a shift in the underlying distribution $P$ of each predictor separately. We measure the distributional stability of $\mathbb{E}_P[\tau(X)]$ while keeping the conditional distribution of the other variables given the predictor constant. The study uses the NSW dataset [29], where the outcome variable is 're78', and the covariates $X$ include 'age', 'education', 'black', 'hispanic', 'married', 'nodegree', and 're75', where 're75' denotes pre-intervention earnings in 1975. We estimate the average treatment effect using augmented inverse probability weighting (AIPW) implemented with causal forests [50], resulting in an estimate of $820$ with a standard deviation of $492$. Our findings show that the $s$-values of average treatment effect conditional on 'age', 'education', 'black', 'hispanic', and 're75' are non-zero. We also note that the average treatment effect is unstable with respect to changes in the marginal distribution of 'age', 'education', and 're75' indicating that the average treatment effect can change its sign with a shift in the marginal distribution of these covariates ($s_X > 0.85$). We present the directional $s$-values in Table 1.

Table 1: S-values for NSW data set

| Feature | Age | Education | Black | Hispanic | Married | Nodegree | Re75 |
|---|---|---|---|---|---|---|---|
| Directional s-value | 0.97 | 0.91 | 0.52 | 0.54 | 0 | 0 | 0.96 |

**Parameter transfer.** In this section, we evaluate the two-stage transfer learning procedure described in Section 4 using the DJW subset of the original Lalonde data extracted by Dehejia and Wahba [14] to generate training and test datasets with different distributions. The remaining samples are referred to as DJWC. The DJW subset includes 185 treated and 260 control observations and has additional information on pre-interventional earnings in 1974. We present the pre-intervention characteristics of the two subsets in Appendix J and find that they differ in distribution along several variables, which are statistically significant. This split into training and test datasets allows us to evaluate the transfer learning method in a setting with strong covariate shift. We estimate the average treatment effect separately in the two subsets using causal forest [50]. The estimate on the DJW subset was 1636.7 with a standard deviation of 668.8, while on DJWC, it was -847.5 with a standard deviation of 657.2.

Next, we obtain our training set by adding some proportion $\alpha$ of randomly chosen samples from the DJW subset to the DJWC subset, where $\alpha$ takes values in the set $0.05, 0.1, 0.2, 0.3$, and use

the remaining samples as the test set. We use the procedure described in Section 4 to obtain a projection of the training distribution that closely approximates the test distribution. We use two transfer methods: full transfer, where we use all the covariates for the transfer, and partial transfer, where we use only the subset of covariates with which the ATE is most unstable, namely 'age', 'education', and 're75'. We display our results in Figure 3, where we find that both transfer methods lead to lower ATE estimation error than the naive procedure. Further, we observe that there is not much gain with the full transfer method that uses all the covariates over the partial transfer method.

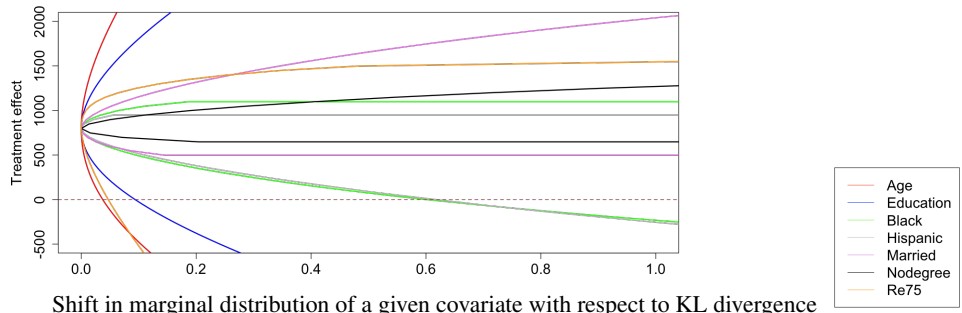

Shift in marginal distribution of a given covariate with respect to KL divergence

Figure 2: The plot shows the estimated minimum and maximum value of the average treatment effect for NSW data achievable when allowing for distribution shift in some covariate (cf. equation (9)).

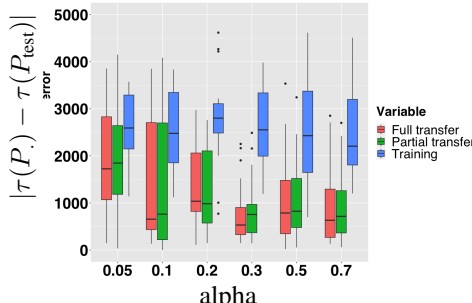

Figure 3: Parameter transfer on the NSW data set. The transfer procedure described in Section 4 compared to a naive procedure that uses only the training distribution, and a full transfer procedure that uses data on all covariates from the new distribution. The green, red and blue bars represent performance of transfer learning with partial, full new data, and naive method respectively. Error bars show the range of error over 20 repetitions.

## 5.2 Wine Quality data set

We evaluate the effectiveness of our method using the wine quality dataset from the UCI Machine Learning Repository [13, 18]. The dataset includes subgroups of red and white wines, each with $11$ chemical properties used as predictors. The response is a continuous quality assessment measured on a scale of $0$ to $10$. The dataset consists of $1599$ red wines and $4898$ white wines. We use all red wines as the training set and randomly select a proportion $\alpha$ from the white wines, where $\alpha \in \{0.01, 0.05, 0.1\}$. The remaining observations are used for testing. We include a small proportion of white wines in the red wine training set to ensure that the shifted distribution is absolutely continuous with respect to the training distribution. This step is necessary to avoid making transfer learning very challenging when datasets deviate from this assumption.

**Distributional stability of regression coefficients.** We use directional $s$-values to assess the distributional stability of ordinary least-squares regression coefficients. We focus our analysis on the predictors "pH" and "density," although similar results can be obtained for other variables. Figure 4 shows estimates of the minimum and maximum achievable value of a regression coefficient given

a specific shift in the marginal distribution of a covariate (as defined in equation (9)). We observe that the coefficient of "pH" is unstable with respect to shifts in "fixed.acidity," "chlorides," "pH," "sulphates," and "alcohol" ($s_X > 0.85$). The coefficient of "density" is unstable with respect to shifts in "volatile.acidity," "total.sulfur.dioxide," "sulphates," and "alcohol" ($s_X > 0.85$). We present the directional $s$-values in Tables 2 and 3.

**Parameter transfer.** We employ Jin and Rothenhäusler [28]'s transfer procedure to estimate the parameter under shifted distributions. This transfer procedure combines a re-weighting step with a bias-correction step for semi-parametrically efficient transfer. We compare three different estimators. The first one uses only the training distribution, the second transfers the parameter using the covariates found to be unstable in the previous step, and the third employs all covariates for transfer (full transfer). Figure 5 displays the estimation error of the parameter under the projected and training distributions. Both transfer learning methods have smaller errors than the naive estimator that uses only the training distribution. However, the full transfer method that employs all covariates does not yield much improvement over partial transfer. For $\alpha = 0.01$, there is not much enhancement in the coefficient of "density," which might be due to a partial violation of the assumption that the test distribution is absolutely continuous with respect to the training distribution.

Table 2: Directional S-values for wine quality data set (parameter "pH")

f.a = fixed.acidity, v.a = volatile.acidity, c.a = citric.acid, f.so2 = free.sulfur.dioxide, t.so2 = total.sulfure.dioxide, cl = chlorides, so4 = sulphates

| Feature | f.a | v.a | c.a | r.s | cl | f.so2 | t.so2 | density | pH | so4 | alcohol |
|---------|------|------|------|------|------|-------|-------|---------|------|------|---------|
| s-value | 0.86 | 0.81 | 0.65 | 0.83 | 0.94 | 0.55 | 0.8 | 0.83 | 0.97 | 0.97 | 0.88 |

Table 3: Directional S-values for wine quality data set (parameter "density")

f.a = fixed.acidity, v.a = volatile.acidity, c.a = citric.acid, f.so2 = free.sulfur.dioxide, t.so2 = total.sulfure.dioxide, cl = chlorides, so4 = sulphates

| Feature | f.a | v.a | c.a | r.s | cl | f.so2 | t.so2 | density | pH | so4 | alcohol |
|---------|------|------|------|------|------|-------|-------|---------|------|------|---------|
| s-value | 0.80 | 0.94 | 0.83 | 0.81 | 0.78 | 0.81 | 0.93 | 0.84 | 0.79 | 0.9 | 0.98 |

# 6 Discussion

The generalizability and replicability of statistical findings are crucial in scientific research. However, classical statistical measures only account for uncertainty due to sampling and not other sources of variation, such as distributional shift. Since distributions are expected to vary between settings and locations, it is essential to understand how statistical parameters are affected by such shifts to assess the stability of a finding.

In this work, we propose stability measures to quantify the impact of distributional shifts on statistical parameters at an overall and variable-specific level, respectively, enabling a more detailed evaluation of instability. We expect that stability measures will be used in tandem with transfer learning procedures. Initially, the stability of a conclusion can be assessed with respect to various shifts, and then, once the sensitivities are determined, the data scientist can collect data from target distributions for the most sensitive covariates and update the model accordingly. As an example, researchers are often interested in a causal effect estimate for a new location. For some covariates such as age and education, partial data is available via surveys such as American National Election Study (ANES) or Cooperative Election Study (CES). Additional partial data can be cheaply obtained via Amazon Mechanical Turk. However, some covariates are hard to collect, since they require running a study in the new location. Our numerical results show that the proposed approach can help prioritize data collection.

The proposed approach has several limitations. First, we consider worst-case shifts, which can be somewhat pessimistic for situations where the distribution is expected to change due to random

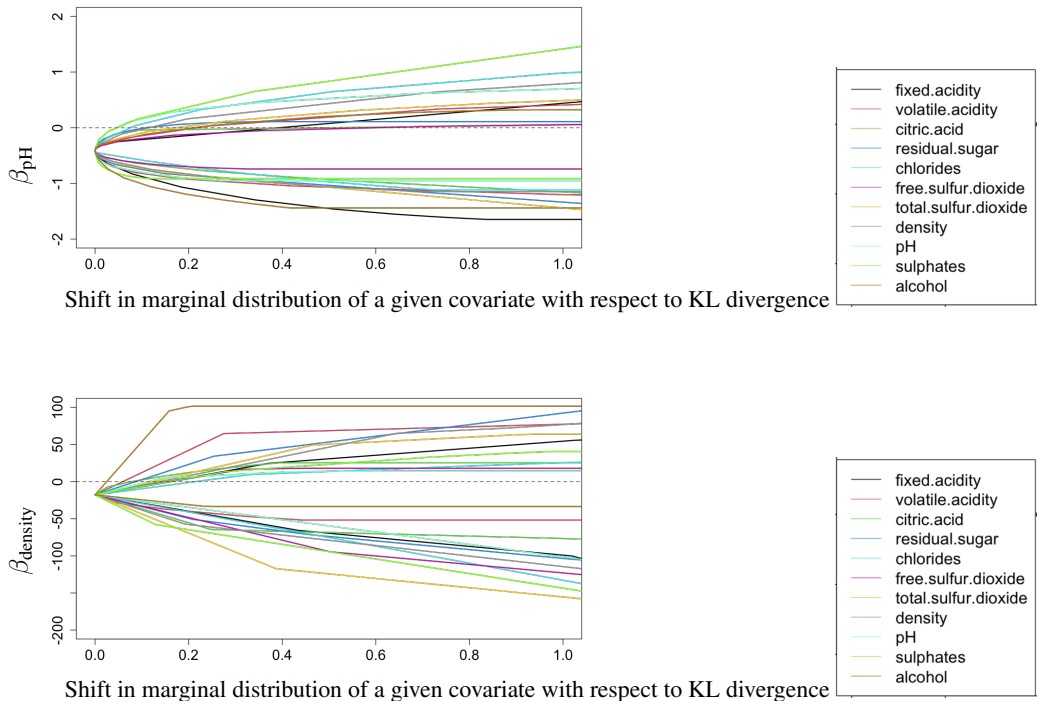

Figure 4: The plot shows the estimated minimum and maximum value of the regression coefficient for wine quality data set achievable under a distribution shift in one covariate (as defined in equation (9)).

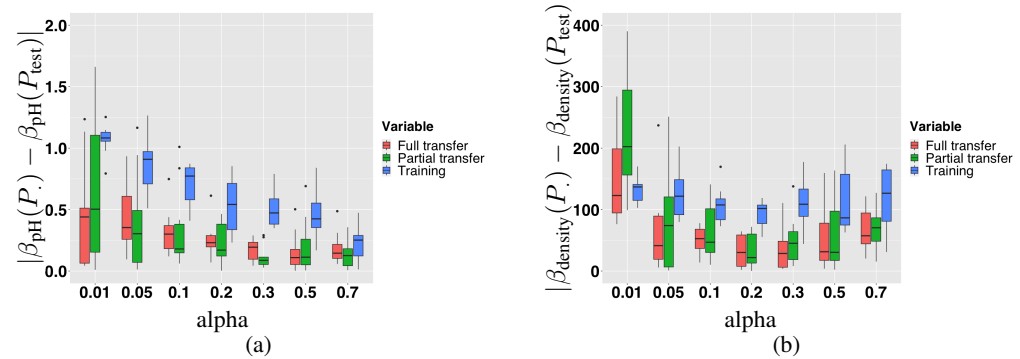

Figure 5: This figure shows the effectiveness of a two-stage transfer procedure for the wine quality data set. The green, red and blue bars represent performance of transfer learning with partial, full new data, and naive method respectively. Error bars show the range of error over 20 repetitions. Transfer learning outperforms the naive method in almost all cases.

perturbations in background characteristics. Modelling distribution shift as random is an attractive alternative [41, 26]. Secondly, the Kullback-Leibler divergence only allows for certain types of distribution shift. The shifted distribution might have a different support than the training distribution. In this case, it might be more appropriate to model the changes with the total variation distance or the Wasserstein distance. Thirdly, in this paper we focus on distribution shift in the covariates $X$. There is evidence that distributions shift not only in the covariates $X$, but also in $Y|X$ [27, 31]. Developing new tools to better understand distributional shifts in $Y|X$ is an exciting research direction.

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
