# 7   Appendix

## A   Considerations for defining stability values

In the following we want to lay out the thought process that has led us to this exact formulation of stability values. To sum it up, our choice of the $s$-value is guided by familiarity, practicality, and flexibility.

One of the advantages of using the KL divergence is that the procedure is a convex optimization problem for linear estimands. There are other considerations from a practical perspective. First, all continuously differentiable $f$-divergences have an equivalent Taylor expansion in a neighborhood around $0$. From this perspective, for small shifts it does not matter which divergence to choose. Compared to other $f$-divergences, one advantage of the KL divergence is that it is widely known in the statistics and ML communities.

Another popular distance is the total variation distance. The KL divergence is less conservative than the total variation distance which would create very drastic distributional changes. To be more precise, for the parameter $\theta(P') = \mathbb{E}_{P'}[X]$ and all distributions $P$, we would have

$$\inf_{P'} \mathrm{TV}(P_0, P') \text{ such that } \mathbb{E}_{P'}[X] = 0.$$

This infimum is zero, that is the parameter is unstable for any $P$. Thus, the stability value defined via total variation distance is too coarse in the sense that infinitely small shifts will already break common parameters of interest. Thus, its usefulness is very limited in practice.

Comparing our choice

$$s(\mu, P_0) = \sup_{P} \exp\{-D_{KL}(P||P_0)\} \quad \text{s.t.} \quad \mathbb{E}_P[Z] = 0,$$

with the potential choice

$$s'(\mu, P_0) = \inf_{P} D_{KL}(P||P_0) \quad \text{s.t.} \quad \mathbb{E}_P[Z] = 0,$$

the latter can lead to extremely unstable optimization procedures if the argmin is far away from $P_0$. Trying to estimate $s'$ could lead to unreliable and misleading stability evaluations. Our choice of transformation guarantees that the regime where estimation of the KL divergence is unstable, leads to $s$ values that are close to each other. In other words, we "compress" regimes in which the $s$-values are hard to estimate.

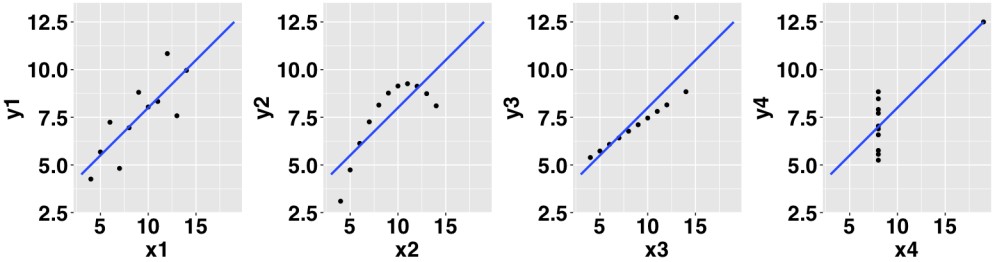

Figure 6: Anscombe's quartet data

## B   Example: Anscombe's quartet

We demonstrate the usage of our method on Anscombe's quartet [2], which comprises of four data sets that yield nearly identical OLS estimates and $p$-values (see Figure 6). While $p$-values cannot unveil the difference in distributional stability of the regression coefficients among the four data sets, the proposed measure captures the stability of the regression coefficients under distribution shift.

In Table 4, we display the OLS estimate, $p$-values and our $s$-values (both general and variable specific). While the $p$-value is the same for all data sets, the $s$-values differ. The regression coefficient

Table 4: This table exhibits the OLS estimate, $p$-values, general and variable-specific $s$-values of the regression coefficient for each data set in Anscombe's quartet. The $p$-values are the same for all data sets. The $s$-values vary drastically across the data sets, indicating that the parameter is more stable under distribution shift for data set 3 and data set 4 than for data set 1 and data set 2.

| Part | | | | |
| $Y = \beta_0 + X\beta_1$ | OLS estimate ($\beta_1$) | $p$-values | $s$ | $s_X$ |
| --- | --- | --- | --- | --- |
| Set 1 | 0.5 | 0.00217 | 0.465 | 0 |
| Set 2 | 0.5 | 0.00217 | 0.63 | 0.63 |
| Set 3 | 0.5 | 0.00217 | 0 | 0 |
| Set 4 | 0.5 | 0.00217 | 0 | 0 |

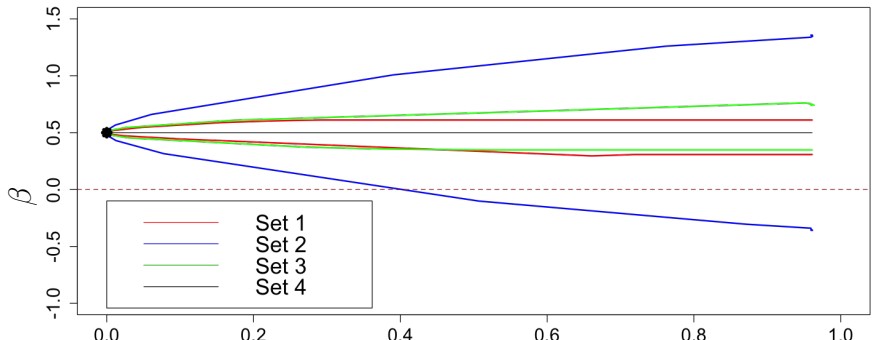

Shift in marginal distribution of the covariate with respect to KL divergence

Figure 7: The plot shows the minimum and maximum value of the regression coefficient ($\beta$) achievable by a shift in the marginal distribution of the covariate $X$. More specifically, the upper and lower bounds are estimated versions of the bounds in equation (9).

in set 1 has a $s$-value of $0.465$, which indicates that the regression coefficient may be null under distributional shifts. However, when considering directional shifts with $E = X$, one obtains the directional $s$-value $s_X = 0$. For Set 2, both types of $s$-values take the same non-zero value while sets 3 and 4 have $s = s_E = 0$. Thus, the proposed stability measure coincides with the intuition that the regression coefficient is relatively stable in data sets 3 and 4 under distribution shift.

In Figure 7, we plot estimated versions of the upper and lower bounds as defined in (9) across $c$ for different choices of the variable $E$.

In this example, distributional instability of regression coefficients mostly occurs due to model misspecification. In practice, we can test for model misspecification using classical approaches like the Ramsey Regression Equation Specification Error Test (RESET) test [35] or via diagnostic tests. However, such tests do not quantify instability in terms of distributional shifts, and distributional instability can occur even if models are well-specified.

## C  Consistency and asymptotic normality

### C.1  $S$-value of the mean

Here we present the consistency results of the estimator of $s$-value of mean $\hat{s}(\mu, P_n)$ defined in equation (5).

**Lemma C.1** (Consistency of $s$-value). *Let $Z \sim P_0$ be a real-valued non-degenerate random variable with mean $\mu(P_0) = \mathbb{E}_{P_0}[Z]$ and a finite moment generating function on $\mathbb{R}$. Then $\hat{s}(\mu, P_n) \xrightarrow{P} s(\mu, P_0)$ as $n \to \infty$.*

We provide the proof of the above lemma in Appendix I.1. Now let us turn to asymptotic normality.

**Lemma C.2** (Asymptotic normality of $s$-values)**.** *Let $Z \sim P_0$ be a real-valued random variable with mean $\mu(P_0) = \mathbb{E}_{P_0}[Z]$ and a finite moment generating function on $\mathbb{R}$. Assume that $\mathbb{E}[Z^2 e^{\lambda^* Z}] > 0$. Let $\hat{\lambda} = \arg\min \frac{1}{n} \sum_{i=1}^n e^{\lambda Z}$ and $\lambda^* = \arg\min \mathbb{E}_{P_0}[e^{\lambda Z}]$. Then,*

$$\frac{1}{n} \sum_{i=1}^n e^{\hat{\lambda} Z} - \mathbb{E}_{P_0}[e^{\lambda^* Z}] \overset{d}{=} \mathcal{N}\left(0, \frac{Var_{P_0}(e^{\lambda^* Z})}{n}\right) + o_P(1/n).$$

The proof for this result can be found in the Appendix, Section H. Based on this result, one can construct confidence intervals for the $s$-value. To be more precise, for fixed $\alpha \in (0, 1)$, one can define confidence intervals via $\hat{s} \pm z_{1-\alpha/2} \frac{\hat{\sigma}}{\sqrt{n}}$, where $\hat{\sigma}^2$ is the empirical variance of $(e^{\hat{\lambda} Z_i})_{i=1,\dots,n}$ and $z_{1-\alpha/2}$ is the $1 - \alpha/2$ quantile of a standard Gaussian random variable.

**Directional $s$-values**

Let us now discuss consistency of $\hat{s}_E(\mu, P_n)$. We make the following regularity assumption.

**Assumption 1.** *Let $\hat{f}_n(\cdot)$ be an estimate of $\mathbb{E}_{P_0}[Z|E = \cdot]$ defined over $\mathcal{E}$. We assume that $\sup_{e \in \mathcal{E}} |\mathbb{E}_{P_0}[Z|E = e] - \hat{f}_n(e)|_\infty \to 0$.*

**Lemma C.3** (Consistency of directional $s$-value)**.** *Under the setting of Theorem 2 and Assumption 1, we have $\hat{s}_E(\theta, P_n) \overset{P}{\to} s_E(\theta, P_0)$ as $n \to \infty$.*

We present the proof of Lemma C.3 in Appendix I.3. To obtain asymptotically valid confidence intervals for $s$-values, one has to deal with the fact that the non-parametric estimates $\hat{f}_n(\cdot)$ may converge very slowly to the ground truth. To deal with this problem, we employ a debiasing technique.

**Lemma C.4** (Asymptotic normality of directional $s$-values)**.** *Let $\hat{f}_n(\cdot)$ be an estimate of $f(\cdot) = \mathbb{E}_{P_0}[Z|E = \cdot]$. We assume that $\hat{f}_n$ is fit on a held-out portion of the data set, that is $\hat{f}_n(\cdot)$ is independent of $D_i, i = 1, \dots, n$. We assume that $\sup_{e \in \mathcal{E}} |\hat{f}_n(e) - f(e)| = o_P(n^{-1/4})$. Furthermore, we assume that the moment generating function of $Z$ is finite on $\mathbb{R}$ and that the matrix $\mathbb{E}_{P_0}[f(E)^2 e^{\lambda^* f(E)}] > 0$. Let $\hat{\lambda} = \arg\min \frac{1}{n} \sum_{i=1}^n e^{\lambda \hat{f}_n(E_i)}$ and $\lambda^* = \arg\min \mathbb{E}_{P_0}[e^{\lambda f(E)}]$. Then,*

$$\frac{1}{n} \sum_{i=1}^n (1 + \hat{\lambda} Z_i - \hat{\lambda} \hat{f}_n(E_i)) e^{\hat{\lambda} \hat{f}_n(E_i)} - \mathbb{E}_{P_0}[e^{(\lambda^*) f(E)}] \overset{d}{=} \mathcal{N}\left(0, \frac{\sigma_s^2}{n}\right) + o_P(1/n),$$

*where*

$$\sigma_s^2 = Var_{P_0}(e^{\lambda^* f(E)}) + Var_{P_0}(e^{\lambda^* f(E)} \lambda^* (Z - f(E))).$$

The proof of this result can be found in the appendix, Section H. One can construct asymptotically valid confidence intervals based on this result, analogously as discussed after Lemma C.2. The de-biasing technique uses sample splitting, which reduces efficiency. Full efficiency can be obtained by using cross-fitting techniques, see for example Chernozhukov et al. [8]. The debiasing technique leads to asymptotically unbiased and normal estimates, but it does come at the cost of stability. This can be easily seen from the formula: the standard approach has asymptotic variance $Var_{P_0}(e^{\lambda^* f(E)})$, which is larger than $\sigma_s^2$ unless $Z \equiv f(E)$. For this reason and for simplicity, in the main paper we stick to the estimator in equation (8), instead of the more unstable debiased estimator.

# D  S-values of parameters defined via risk minimization

Here, we discuss how to compute $s$-values for parameters defined via risk minimization and generalize to multi-parameter settings. Let us describe two examples where this is of interest. First, for a parameter vector $\eta$, each component might correspond to the causal effect of a subgroup. One may then ask whether there is a small distribution shift that renders all causal effects zero. The theory outlined in this section is also relevant for settings where one is interested in a single parameter in the presence of nuisance parameters. For example, in causal inference one component of the parameter vector might corresponds to the causal effect of interest, while the other components of the vector might correspond to the effect of observed confounders, which are not of interest by themselves.

We consider the following setting. Let $\Theta \subseteq \mathbb{R}^p$ be the parameter (model) space, $P_0$ be the data generating distribution (training distribution) on the measure space $(\mathcal{Z}, \mathcal{A})$, $Z$ be a random element of $\mathcal{Z}$, and $L : \Theta \times \mathcal{Z} \to \mathbb{R}$ be a loss function, which is strictly convex and differentiable in its first argument. Define the parameter $\theta^M(P)$ for $P \in \mathcal{P}$ via

$$\theta^M(P) = \arg\min_{\theta \in \Theta} \mathbb{E}_P[L(\theta, Z)]. \tag{10}$$

Let $\ell(\theta, Z) = \partial_\theta L(\theta, Z)$. So far, for simplicity we have only considered $s$-values of one-dimensional parameters. In practice, we need a slightly more general notion of $s$-values that can handle $p$-dimensional parameters. For $\eta \in \mathbb{R}^p$, define the extended $s$-value via

$$s(\theta^M - \eta, P_0) = \sup_{P \in \mathcal{P}} \exp\{-D_{KL}(P||P_0)\} \quad \text{s.t.} \quad \theta^M(P) - \eta = 0. \tag{11}$$

Choosing $\eta$ is similar to choosing a null hypothesis for significance testing in statistical decision problems. For example, in linear regression it is common to test the global null, where the regression coefficient is assumed to be zero for all components. Analogously, in this setting one might ask whether just a small distributional shift can shift all components of the parameter to zero.

Similar to the one-dimensional mean case (Section 3.1), the $s$-value in (11) can be obtained by solving a $p$-dimensional convex optimization problem that we state in the following corollary. Its proof can be found in Appendix I.4.

**Corollary D.1.** *Let $\ell(\eta, Z)$ have a finite moment generating function on $\mathbb{R}^p$ under $P_0$ for all $\eta \in \Theta$. Then, the $s$-value of the parameter $\theta^M$ as defined in (10) is given by*

$$s(\theta^M - \eta, P_0) = \inf_{\lambda \in \mathbb{R}^p} E_{P_0}[e^{\lambda^\intercal \ell(\eta, Z)}]. \tag{12}$$

Consistency and asymptotic normality can be obtained analogously as in Lemma C.1 and Lemma C.2. A general theorem is formulated in the Appendix, Section H. We next present some examples of parameters defined via risk minimization.

**Example 5** (Regression). *Let $\mathcal{X} \subseteq \mathbb{R}^p$ be a $p$-dimensional feature space and $\mathcal{Y}$ be the space of response. Let $Y \in \mathcal{Y}$ satisfy $Y = X\theta + \epsilon$, where $\theta \in \Theta \subset \mathbb{R}^p$ and $\epsilon$ is uncorrelated with $X$. Then the OLS parameter $\theta^M(P)$ is given by*

$$\theta^M(P) = \operatorname*{argmin}_\theta \mathbb{E}_P[(Y - X\theta)^2].$$

*If $X^\intercal(Y - X\eta)$ has finite moment generating function on $\mathbb{R}^p$ for all $\eta \in \Theta$ then using Corollary D.1, we have*

$$s(\theta^M - \eta, P_0) = \inf_{\lambda \in \mathbb{R}^p} \mathbb{E}_{P_0}[e^{\lambda^\intercal X^\intercal(Y - X\eta)}].$$

**Example 6** (Generalized linear models). *Let $\mathcal{X} \subseteq \mathbb{R}^p$ be a $p$-dimensional feature space and $\mathcal{Y}$ be the space of response. Let $Y \in \mathcal{Y}$ satisfy $\mathbb{E}[Y \mid X] = g^{-1}(X\theta)$ where $\theta \in \Theta \subset \mathbb{R}^p$ and $g$ is the link function. With slight abuse of notation, let $L(Y, X\theta)$ be the negative log-likelihood function. The maximum likelihood parameter $\theta^M$ is given by*

$$\theta^M(P) = \operatorname*{argmin}_\theta \mathbb{E}_P[L(Y, X\theta)].$$

*If $X^\intercal \partial_2 L(Y, X\eta)$ has finite moment generating function on $\mathbb{R}^p$ for all $\eta \in \Theta$ then using Corollary D.1, we have*

$$s(\theta^M - \eta, P_0) = \inf_{\lambda \in \mathbb{R}^p} \mathbb{E}_{P_0}[e^{\lambda^\intercal X^\intercal \partial_2 L(Y, X\eta)}].$$

We next characterize directional $s$-values. We define the extended directional $s$-value as

$$s_E(\theta^M - \eta, P_0) = \sup_{P \in \mathcal{P}, P[\bullet|E] = P_0[\bullet|E]} \exp\{-D_{KL}(P||P_0)\} \quad \text{s.t.} \quad \theta^M(P) - \eta = 0. \tag{13}$$

Similarly as above, directional $s$-values can be obtained by solving a convex optimization problem that we state in the following corollary. We present the proof in Appendix I.5.

**Corollary D.2** (Directional shifts). *Let $\ell(\eta, Z)$ have a finite moment generating function on $\mathbb{R}^p$ under $P_0$. Then,*

$$s_E(\theta^M - \eta, P_0) = \inf_\lambda \mathbb{E}_{P_0}[e^{\lambda^\intercal E_{P_0}[\ell(Z, \eta)|E]}]. \tag{14}$$

Consistency and asymptotic normality can be obtained analogously as in Lemma C.3 and Lemma C.4. A general theorem is formulated in the Appendix, Section H.

Next, we present an example from regression setting.

**Example 7** (Regression). *Let $\mathcal{X} \subseteq \mathbb{R}^p$ be a p-dimensional feature space and $\mathcal{Y}$ be the space of the response. Let $Y \in \mathcal{Y}$ satisfy $Y = X\theta + \epsilon$, where $\theta \in \Theta \subset \mathbb{R}^p$ and $\epsilon$ is independent of $X$. As above, the OLS parameter $\theta^M(P)$ is defined as*

$$\theta^M(P) = \underset{\theta}{\operatorname{argmin}} \, \mathbb{E}_P[(Y - X\theta)^2].$$

*If $X^\intercal(Y - X\eta)$ has finite moment generating function on $\mathbb{R}^p$ for all $\eta \in \Theta$, then Corollary D.2 implies*

$$s_X(\theta^M - \eta, P_0) = \inf_{\lambda \in \mathbb{R}^p} \mathbb{E}_{P_0}[e^{\lambda^\intercal X^\intercal(\mathbb{E}_{P_0}[Y|X] - X\eta)}].$$

*Now let us investigate the case with high directional distributional stability with respect to $E = X$. In the following, let us assume that $s_X(\theta^M - \eta, P_0) = 0$ for all $\eta \neq \theta(P_0)$ and that $X$ has positive density with respect to the Lebesgue measure. By definition of the s-value we have $\theta^M(P) = \theta^M(P_0)$ for every measure $P$ that is absolutely continuous with respect to $P_0$ and satisfies $P(\cdot|X = x) = P_0(\cdot|X = x)$ for all $x \in \mathcal{X}$. By definition of OLS, we must have that almost surely*

$$\mathbb{E}_{P_0}[X^\intercal(Y - X\theta^M)|X] = 0.$$

*Thus, almost surely,*

$$X^\intercal(\mathbb{E}_{P_0}[Y|X] - X\theta^M) = \mathbb{E}_{P_0}[X^\intercal(Y - X\theta^M)|X] = 0.$$

*If $X$ has a density with respect to the Lebesgue measure, then $\mathbb{E}_{P_0}[Y|X] = X\theta^M$ almost surely. Thus, directional distributional instability in linear models with respect to $E = X$ is related to whether the linear model is a good approximation of the regression surface, i.e. $\mathbb{E}_{P_0}[Y|X] \approx X\theta^M$. More specifically, if the linear model is a good approximation of the regression surface, directional stability is high. This is an example, where distributional instability can be induced by model misspecification. As discussed earlier in Section B, distributional instability can also be induced by other sources such as the presence of exogenous covariates that are correlated both with covariates and the outcome.*

We can similarly obtain results for generalized linear models (see Example 6).

## D.1 S-values for a single component

In many cases, practitioners may be interested in obtaining the $s$-value for a single component of $\theta^M \in \mathbb{R}^p$ instead of the entire vector $\theta^M$. For example, in causal inference, one component of $\theta^M$ can correspond to average treatment effect while other parameters may not be of scientific interest. In such settings, one might want to evaluate the stability of the parameter of interest (the average treatment effect) and not the stability of the nuisance components. Let $\theta_k^M$ be the $k$-th component of parameter vector $\theta^M \in \mathbb{R}^p$ for $k \in \{1, \ldots, p\}$.

Intuitively, to obtain the $s$-value of a single parameter, we seek for the smallest possible shift in distribution that tilts the parameter to a pre-determined value, hence, we take supremum over other remaining nuissance parameters. To be more precise, using the definition of $s$-values and Corollary D.1, we have

$$
\begin{aligned}
s(\theta_k^M - \eta_k, P_0) &= \sup_{P \in \mathcal{P}} \exp\{-D_{KL}(P||P_0)\} \text{ s.t. } \eta_k(P) - \eta_k = 0 \\
&= \sup_{\eta_1, \ldots, \eta_{k-1}, \eta_{k+1}, \ldots, \eta_p} \sup_{P \in \mathcal{P}} \exp\{-D_{KL}(P||P_0)\} \text{ s.t. } \eta(P) - \eta = 0 \\
&= \sup_{\eta_1, \ldots, \eta_{k-1}, \eta_{k+1}, \ldots, \eta_p} \inf_{\lambda \in \mathbb{R}^p} \mathbb{E}_{P_0}[e^{\lambda^\intercal \ell(\eta, Z)}].
\end{aligned}
\tag{15}
$$

We next obtain a finite sample estimate of $s$-values for individual components of $\theta^M$ and show that it is consistent to the population version. Let $P_n$ be the empirical distribution of $Z_i \overset{\text{i.i.d.}}{\sim} P_0$. We propose to estimate the $s$-value via the plugin estimator

$$\hat{s}(\theta_k^M - \eta_k, P_n) = \sup_{\eta_1,\ldots,\eta_{k-1},\eta_{k+1},\ldots,\eta_p} \inf_{\lambda \in \mathbb{R}^p} \mathbb{E}_{P_n}[e^{\lambda^\intercal \ell(\eta,Z)}]$$

$$= \sup_{\eta_1,\ldots,\eta_{k-1},\eta_{k+1},\ldots,\eta_p} \inf_{\lambda \in \mathbb{R}^p} \frac{1}{n}\sum_{i=1}^n e^{\lambda^\intercal \ell(\eta,Z_i)} \tag{16}$$

This is a challenging optimization problem. The optimization problem in (16) is non-convex and hence, finding the global optimum in practice is intractable. In Appendix F, we propose algorithms to solve the optimization problem in (16) with convergence guarantees. We also propose a simple plug-in estimate in Appendix E.

The non-convexity of the optimization problem has important consequences for interpreting $s$-values. For the sake of simplicity, let us assume that $n$ is large, that is that estimation errors due to finite samples can be ignored. If the estimated $s$-value is large, the algorithm constructs a small distribution shift that changes the sign of the parameter. In other words, the parameter is certifiably unstable. On the other hand, if the estimated $s$-value is small, then this could be due the algorithm being stuck in a local optimum. Thus, for non-convex problems a large $s$-value is an indication of instability, but a small $s$-value cannot be necessarily interpreted as a proof of stability. This is similar to hypothesis testing in statistical inference, where the non-significance of a $p$-value cannot be interpreted as a proof of the null hypothesis.

We next show consistency of $\hat{s}(\theta_k^M - \eta_k, P_n)$ to the corresponding population stability value $s(\theta_k^M - \eta_k, P_0)$. To this end, we make the following assumption.

**Assumption 2.** *Let $\Sigma \subset \mathbb{R}^p$ be a compact subset such that the map $\eta \to \ell(\eta, Z)$ is continuous on $\Sigma$ and $\mathbb{E}_{P_0}[\sup_{\eta \in \Sigma} e^{\lambda^\intercal \ell(\eta,Z)}] < \infty$ for any $\lambda \in \mathbb{R}^p$.*

**Lemma D.1** (Consistency of $s$-value). *Let $\Sigma_k$ denote the projection of $\Sigma$ on the $k$th co-ordinate. Under Assumption 2, we have $\sup_{\eta_k \in \Sigma_k} |\hat{s}(\theta_k^M - \eta_k, P_n) - s(\theta_k^M - \eta_k, P_0)| \xrightarrow{P} 0$ for $k = \{1, \ldots, p\}$ as $n \to \infty$.*

We present the proof of Lemma D.1 in Appendix I.6.

**Example 8** (Regression). *Let $\mathcal{X} \subseteq \mathbb{R}^p$ be a $p$-dimensional bounded feature space and $\mathcal{Y}$ be the space of the response. Let $Y \in \mathcal{Y}$ satisfy $Y = X^\intercal \beta + \epsilon$, where $\beta \in \Theta \subset \mathbb{R}^p$ and $\epsilon$ is independent of $X$. We have $L(\eta, X, Y) = \frac{1}{2}(Y - X^\intercal \eta)^2$, $\ell(\eta, X, Y) = -X(Y - X^\intercal \eta)$ and hence, $\nabla_\eta \ell(\eta, X, Y) = XX^T$. Now, for a compact subset $\Sigma \subset \mathbb{R}$, $\mathbb{E}_{P_0}[\sup_{\eta \in \Sigma} e^{\lambda^\intercal \ell(\eta,Z)}] < \infty$ if $\epsilon$ has a finite moment generating function on $\mathbb{R}$. Invoking Lemma D.1, we can conclude that the estimator is consistent.*

*Let us now discuss how to estimate directional $s$-values of individual components. Suppose we want to obtain the directional $s$-value of the $k$-th component of vector $\theta^M \in \mathbb{R}^p$. Using Corollary D.2, the population directional $s$-value is given by*

$$s_E(\theta_k^M - \eta_k, P_0) = \sup_{\eta_1,\ldots,\eta_{k-1},\eta_{k+1},\ldots,\eta_p} \inf_{\lambda \in \mathbb{R}^p} E_{P_0}[e^{\lambda^\intercal \mathbb{E}_{P_0}[\ell(\eta,Z)|E]}]. \tag{17}$$

Now we propose a finite sample estimator of the directional $s$-value of individual components and show consistency.

Let $Q_n(\eta, E)$ be a finite sample estimator of $\mathbb{E}_{P_0}[\ell(\eta, Z) \mid E]$, then the finite sample plugin estimator is given by

$$\hat{s}_E(\theta_k^M - \eta_k, P_n) = \sup_{\eta_1,\ldots,\eta_{k-1},\eta_{k+1},\ldots,\eta_p} \inf_{\lambda \in \mathbb{R}^p} E_{P_n}[e^{\lambda^\intercal Q_n(\eta,E)}]. \tag{18}$$

Again, this is a challenging optimization problem. We discuss algorithms in Appendix G. A simplification that allows to derive upper bounds can be found in Appendix, Section E. We make the following additional assumption to show consistency of $\hat{s}_E(\theta_k^M - \eta_k, P_n)$.

**Assumption 3.** *$\sup_\eta \sup_e \|E_{P_0}[\ell(\eta, Z)|E = e] - Q_n(\eta, e)\|_\infty \to 0$, where $Q_n(\eta, e)$ is an estimate of $E_{P_0}[\ell(\eta, Z)|E = e]$.*

**Lemma D.2** (Consistency of directional $s$-value). *Under Assumptions 2 and 3, we have*

$$\sup_{\eta_k \in \Sigma_k} |\hat{s}_E(\theta_k^M - \eta_k, P_n) - s_E(\theta_k^M - \eta_k, P_0)| \xrightarrow{P} 0 \text{ for } k = \{1, \ldots, p\}$$

*as $n \to \infty$.*

We present the proof of Lemma D.2 in Appendix I.7. To quantify uncertainty, in this settings we find the bootstrap preferable over asymptotic expansions (as in Section 3) to account for the fact that the algorithm might get stuck in a local minimum.

# E   A simple plug-in estimator

Equation (15) and equation (17) are non-convex optimization problems that are potentially difficult to solve. In practice, we can obtain a lower bound by removing the outer supremum in (15) and (17) and using a plug-in estimator for the lower bound.

Let $\tilde{\eta} = (\hat{\theta}_1^M, \ldots, \hat{\theta}_{k-1}^M, \eta_k, \hat{\theta}_{k+1}^M, \ldots, \hat{\theta}_p^M)$, where the $\hat{\theta}_i^M$ are estimates of $\theta_i^M(P_0)$. Furthermore, let $Q_n(\eta, E)$ be an estimate of $\mathbb{E}[\ell(\eta, Z)|E]$. We can obtain plug-in estimators of $s(\theta_k^M - \eta_k, P_0)$ and $s_E(\theta_k^M - \eta_k, P_0)$ via

$$\hat{s}_{\text{plug-in}}(\theta_k^M - \eta_k, P_0) = \inf_{\lambda \in \mathbb{R}^p} \mathbb{E}_{P_n}[e^{\lambda^\intercal \ell(\tilde{\eta}, Z)}]$$

$$= \inf_{\lambda \in \mathbb{R}^p} \frac{1}{n} \sum_{i=1}^n e^{\lambda^\intercal \ell(\tilde{\eta}, Z_i)} \text{ and}$$

$$\hat{s}_{E,\text{plug-in}}(\theta_k^M - \eta_k, P_0) = \inf_{\lambda \in \mathbb{R}^p} E_{P_n}[e^{\lambda^\intercal \mathbb{E}_{P_0}[\ell(\tilde{\eta}, Z)|E]}]$$

$$= \inf_{\lambda \in \mathbb{R}^p} \frac{1}{n} \sum_{i=1}^n e^{\lambda^\intercal Q_n(\tilde{\eta}, E_i)}.$$

Clearly, the objective functions are convex and hence the optimization problem is easily solvable. Large plug-in estimate certify instability of parameters. However, since these plug-in estimators are based on lower bounds of $s(\theta_k^M - \eta_k, P_0)$ and $s_E(\theta_k^M - \eta_k, P_0)$, estimates close to zero do not certify stability. Overall, the plug-in estimator can be used as a first check to evaluate distributional instability of a parameter.

# F   S-values of general estimands

Here, we are interested in obtaining $s$-values of individual components of parameters defined via risk minimization as in (15). The corresponding optimization problem to obtain $s$-value as in (15) is generally non-convex. Hence, obtaining a globally optimal solution of the optimization problem is very challenging. Here, we characterize the form of a locally optimal solution of the corresponding optimization problem and give algorithms to solve such problems in Appendix F.1. Here we use the original definition of $s$-value as opposed to the form given in (15), that is,

$$s(\theta_k^M, P_0) = \sup_{P \in \mathcal{P}} \exp\{-D_{KL}(P||P_0)\} \quad \text{s.t.} \quad \theta_k^M(P) = 0, \tag{19}$$

For ease of presentation, from here on we denote the parameter of interest as $\theta$ instead of $\theta_k^M$ and consider a finite sample setting where we observe $n$ samples $\{Z_i\}_{i=1}^n \overset{\text{i.i.d.}}{\sim} P_0$ for some distribution $P_0 \in \mathcal{P}$. Let the empirical distribution of $\{Z_i\}_{i=1}^n$ be denoted by $P_{0,n} = \sum_{i=1}^n \frac{1}{n}\delta_i$, where $\delta_i$ is a dirac measure on $Z_i$. Let $W_n = [0,1]^n$ be $n$ dimensional unit cube and let $S_n = \{w \in \mathbb{R}^n : w_1 + \ldots + w_n = 1, w_i \geq 0 \text{ for } i = 1, \ldots, n\}$ be $n$ dimensional probability simplex. We focus on a one dimensional parameter $\theta : S_n \to \mathbb{R}$ where we define for $w \in S_n$, $\theta(w)$ as $\theta(\sum_{i=1}^n w_i \delta_i)$. With a slight abuse of notation from now on, we redefine $\theta$ on the $n$ dimensional unit cube $W_n$ as $\theta(w) = \theta\left(\frac{\sum_{i=1}^n w_i \delta_i}{\sum_i w_i}\right)$ for $w \in W_n$. We recall that we want to obtain (extended) $s$-value of parameter $\theta$ given by

$$s(\theta - c, P_{0,n}) = \sup_{w \in W_n} \exp\{-\sum_{i=1}^n w_i \log(nw_i)\} \quad \text{s.t.} \quad \theta(w) = c, \ \sum_{i=1}^n w_i = 1. \tag{20}$$

where $c$ is a real constant.

The above optimization problem belongs to the class of general constrained minimization problems with equality constraints (see Chapter 3 of Bertsekas [5]). In the following, we present necessary and sufficient conditions for a point to be a local optimum of problem in (20). This characterization can be used to verify if we obtained a locally optimal solution of our optimization problem (20). We first define a locally optimal solution to the problem in (20).

**Definition F.1.** *An element $w^* \in S_n$ (the $n$ dimensional probability simplex) is said to be a locally optimal solution to problem* (20) *if $\theta(w^*) = c$ and there exists a small $\epsilon > 0$ such that $\sum_{i=1}^n w_i^* \log n w_i^* \leq \sum_{i=1}^n w_i \log n w_i$ for all $w \in S_n : \theta(w) = c$ and $\|w - w^*\| < \epsilon$.*

We next present a necessary condition for a point to be a local optimum of (20) that follows immediately from Proposition 3.1.1 of Bertsekas [5].

**Corollary F.1** (Necessary conditions)**.** *Assume that $\theta : \text{int}(W_n) \to \mathbb{R}$ is continuously differentiable. Let $w^* \in S_n$ be a locally optimal solution to problem* (20)*, and assume that there does not exist a constant $r \in \mathbb{R}$ such that $\nabla_w \theta(w^*) = r(1, \ldots, 1)$. Then there exists a constant $\lambda \in \mathbb{R}$ such that*

$$w_i^* \propto e^{\lambda \nabla_i \theta(w^*)} \text{ for all } i = \{1, \ldots, n\}. \tag{21}$$

We next present a sufficient condition for a point to be local optimum of (20) that follows from Proposition 3.2.1 of Bertsekas [5]. To that end, we introduce the Lagrangian function $h : \mathbb{R}^n \times \mathbb{R} \times \mathbb{R} \to \mathbb{R}$ that we define as

$$h(w, \delta, \mu) = \sum_{i=1}^n w_i \log(w_i) + \delta(\theta(w) - c) + \mu(\sum_{i=1}^n w_i - 1) \text{ for } w \in W_n, \text{ and } \delta, \mu \in \mathbb{R}. \tag{22}$$

**Corollary F.2** (Second order sufficiency conditions)**.** *Assume that $\theta : \text{int}(W_n) \to \mathbb{R}$ is twice continuously differentiable, and let $w^* \in W_n$ and $\delta^*, \mu^* \in \mathbb{R}$ satisfy*

$$\nabla_w h(w^*, \delta^*, \mu^*) = 0, \ \nabla_{\delta, \mu} h(w^*, \delta^*, \mu^*) = 0,$$

$$\gamma' \nabla^2_{ww} h(w^*, \delta^*, \mu^*) \gamma > 0, \text{ for all } \gamma \neq 0 \text{ with } \nabla \theta(w^*)' \gamma = 0 \text{ and } \sum_{i=1}^n \gamma_i = 0.$$

*Then $w^*$ is a strict local optimum of* (20)*.*

Based on the characterization of local optima above, we present a Majorization-Minimization based algorithm [28] in Appendix F.1 to solve (20) and give sufficient conditions under which the iterates of the algorithm converges to a point that satisfies the first-order necessary conditions (21). We have similar characterization of locally optimal solution for the optimization problem involved in obtaining the directional $s$-values (2) that we present in Appendix G.1 along with the algorithm to solve such problems.

### F.1 Algorithms to obtain $s$-values for general estimands

Here we present a Majorization-Minimization (MM) based algorithm [28] to solve the problem in (20) and show that it converges to a point that satisfies first-order necessary conditions (21). We also adapt our procedure to obtain directional or variable specific $s$-value. We can use several existing algorithms to solve (20) (see Chapter 4 of Bertsekas [5]) that come with some convergence guarantees. However, the convergence guarantees of the existing algorithms typically come under the assumption that the iterates obtained by the algorithm converge (or we only have guarantees along a subsequence) whereas we present sufficient conditions under which the iterates obtained by our algorithm always converge to a point that satisfies first-order necessary conditions. Further, the existing algorithms require obtaining close approximations to the first-order stationary points of the corresponding augmented Lagrangian (for example, augmenting the objective function with a square of the parameter $\theta$ with a high penalty), however, standard approaches for obtaining first-order stationary points of such functions require slightly stronger assumptions (M-smoothness of the square of $\theta$, see Assumption 4 and the following remark below). Further, since the constraint function involves a one-dimensional parameter $\theta$, our procedure can be efficiently adapted to obtain $s$-value over a range of constants $c$ as in equation (20).

To that end, we make the following smoothness assumption of our parameter $\theta$.

**Assumption 4.** *The function $\theta : int(W_n) \to \mathbb{R}$ is continuously differentiable and $M$ smooth for some $M \in \mathbb{R}$, that is, for $w, w' \in W_n$,*

$$|\theta(w') - \theta(w) - \langle \nabla\theta(w), w' - w \rangle| \leq \frac{M}{2} \|w - w'\|_2^2. \tag{23}$$

Since $\ell_1$ and $\ell_2$ norms are equivalent in finite dimensional spaces, by Pinsker's inequality, we have the following relation for any $w, w' \in S_n$ for some real constant $L > 0$,

$$|\theta(w') - \theta(w) - \langle \nabla\theta(w), w' - w \rangle| \leq L \sum_{i=1}^{n} w_i' \log \frac{w_i'}{w_i}. \tag{24}$$

This new upper bound would help obtain a closed-form expression of update in each iteration of the algorithm (see Proposition 1).

**Remark** In practice, the constant $L$ is often not known in which case, we need to tune it similarly as we would tune the step size in a gradient descent-based method.

**Remark** Although, it appears we make stronger smoothness assumptions for $\theta$ than what is needed for convergence guarantees of algorithms in Chapter 4 of Bertsekas [5], however, such assumptions are standard for convergence guarantees of gradient descent-based methods that are typically used to minimize the augmented Lagrangian at each iterate of the algorithm as needed for example, in Proposition 4.2.2 of Bertsekas [5] where we need $M$-smoothness of the square of $\theta$.

To obtain a solution of (20), we solve the Lagrangian form of the optimization problem in (20) as given by

$$\underset{w_1,\ldots,w_n, w_i \geq 0, \sum w_i = 1}{\text{minimize}} g(w) = \delta(\theta(w) - c) + \sum_{i=1}^{n} w_i \log(w_i) \tag{25}$$

for any fixed $\delta$. If there exists a $\delta$ such that the iterates obtained by our algorithm converges to some $w_{opt}^{\delta} \in W_n$ that satisfies $\theta(w_{opt}^{\delta}) = c$, then we can show that $w_{opt}^{\delta}$ satisfies first order necessary conditions (21). In practice, we use grid search to obtain a $\delta$ that yields $\theta(w_{opt}^{\delta}) = c$.

Now we present the MM based algorithm to solve (25) for a given $\delta$ and parameters that satisfy Assumption 4. Without loss of generality, we assume that $\theta(P_{0,n}) > c$ and hence, we choose $\delta > 0$. First, we upper bound the objective in (25) using inequality (24) so that we have for $w, w' \in W_n$

$$g(w') \leq G_L(w', w) := \delta \left( \theta(w) - c + \langle \nabla\theta(w), w' - w \rangle + L \sum_{i=1}^{n} w_i' \log \frac{w_i'}{w_i} \right) + \sum_{i=1}^{n} w_i' \log w_i'. \tag{26}$$

Note that $G_L(w', w)$ is convex in $w'$ and that $g(w') = G_L(w', w)$ when $w' = w$. Our algorithm then runs iteratively where given a current solution $w^k$, we obtain the next iterate $w^{k+1}$ as $w^{k+1} = \text{argmin}_{w:\sum_{i=1}^{n} w_i = 1} G_L(w, w^k)$, which has a closed form that we present in the proposition below. We define the iteration map $M : W_n \to W_n$ as $M(w^k) = w^{k+1}$ for all $w_k \in W_n$.

**Proposition 1.** *Let $w^{k+1}$ be the iterate obtained at $k$th iteration, that is,*

$$M(w^k) = w^{k+1} = \underset{w:\sum_{i=1}^{n} w_i = 1}{\text{argmin}} \delta \left( \theta(w^k) - c + \langle \nabla\theta(w^k), w - w^k \rangle + L \sum_{i=1}^{n} w_i \log \frac{w_i}{w_i^k} \right) + \sum_{i=1}^{n} w_i \log w_i, \tag{27}$$

*then it is uniquely given by*

$$(M(w^k))_i = w_i^{k+1} \propto e^{-\frac{\delta}{1+L\delta} \nabla_i \theta(w^k)} (w_i^k)^{\frac{L\delta}{1+L\delta}}, \tag{28}$$

*for all $i = \{1, \ldots, n\}$.*

## F.2 Proof of Proposition 1

*Proof.* The optimization problem in (27) is a convex optimization problem and we obtain the solution to (27) via a Lagrange multipliers.

The Lagrangian is given by

$$\operatorname*{argmin}_{w:\sum_{i=1}^{n} w_i = 1} \delta \left( \langle \nabla\theta(w^k), w - w^k \rangle + L \sum_{i=1}^{n} w_i \log \frac{w_i}{w_i^k} \right) + \sum_{i=1}^{n} w_i \log w_i + \gamma(\sum_{i=1}^{n} w_i - 1). \quad (29)$$

Differentiating with respect to $w_i$ and setting the derivative to 0 gives,

$$\delta \nabla_i \theta(w^k) + \delta L \log \frac{w_i}{w_i^k} + \delta L + \log w_i + 1 + \gamma = 0. \quad (30)$$

Hence, the result follows after rearranging the terms and using the constraint $\sum_{i=1}^{n} w_i = 1$. $\qquad \square$

Below we summarize our algorithm to solve (25) for a fixed $\delta$.

---

**Algorithm 1:** Solving (25) for a fixed $\delta$.

**Input:** Training distribution $P_{0,n}$, parameter $\theta$ satisfying Assumption 4 where without loss of generality $\theta(P_{0,n}) > c$, penalty $\delta > 0$, convergence tolerance $\epsilon$ .
**Output:** First order stationary solution of (25).
Set $k \leftarrow 0$, initialize $w^0$ with some $w \in W$, for example, $w_i^0 = \frac{1}{n}$ for all $i = \{1, \dots, n\}$.

1. For $k \geq 0$, obtain $w^{k+1}$ as in (28).

2. Set $k \leftarrow k + 1$.

3. Stop if $g(w^{k+1}) - g(w^k) \leq \epsilon$.

Return $w_{opt}^\delta = w^{k+1}$.

---

We next present the convergence analysis of Algorithm 1 in the following proposition that we prove in Section F.3. First, we recall the definition of a stationary point of a constrained optimization problem where the constraint set is convex.

**Definition F.2.** *Consider the following optimization problem*

$$\operatorname*{minimize}_{x:x\in C} f(x) \quad (31)$$

*where $f : \mathbb{R}^p \to \mathbb{R}$ is differentiable but possibly non-convex, $C \subset \mathbb{R}^p$ is a closed convex set. We call $x^*$ a stationary point of (31) if and only if*

$$\langle \nabla f(x^*), (x - x^*) \rangle \geq 0 \text{ for all } x \in C. \quad (32)$$

**Proposition 2.** *Let $\{w^k\}_{k \geq 1}$ be the sequence of probability distributions generated by Algorithm 1, which solves (25) for some fixed $\delta$ and convergence tolerance $\epsilon = 0$. If there exists a constant $A$ such that $|\theta(w)| \leq A$ for all $w \in W_n$, the unit cube in $n$-dimension, then we have:*

1. *The sequence $\{g(w^k)\}_{k \geq 1}$ is decreasing and converges.*

2. *In addition if all stationary points of (25) are isolated, then the sequence $\{w^k\}_{k \geq 1}$ converges and if $\lim_{k\to\infty} w^k = w_\delta^* \neq (\frac{1}{n}, \dots, \frac{1}{n})$, then $w_\delta^*$ satisfies first order necessary conditions (21), where the constraint in (20) is replaced with $\theta(w) = \theta(w_\delta^*)$.*

Next we use grid search to find $\delta$ (typically increase the value of $\delta$) such that $w_{opt}^\delta$ satisfies $\theta(w_{opt}^\delta) = c$. Below we summarize the algorithm to find a solution of (20) that satisfies first order necessary conditions (21).

---

**Algorithm 2:** $s$-value for general estimands.

---

**Input:** Training distribution $P_0$, parameter $\theta$ satisfying Assumption 4, convergence tolerance $\epsilon$.

**Output:** First order stationary point of (20).

Set $k \leftarrow 1$, initialize $\delta_0 = 0$, $\delta_1 = 2\gamma$ for some small $\gamma > 0$.

1. Run Algorithm 1 with $\delta = \delta_k$ and obtain the output of Algorithm 1 as $w_{opt}^{\delta_k}$.

2. If $|\theta(w_{opt}^{\delta_k}) - c| \leq \epsilon$, stop and return $s(\theta - c, P_{0,n}) = e^{-\sum_{i=1}^{n}(w_{opt}^{\delta_k})_i \log n (w_{opt}^{\delta_k})_i}$.

3. If $\theta(w_{opt}^{\delta_k}) > c + \epsilon$, set $\delta_{k+1} = 2\delta_k$, set $k \leftarrow k + 1$. and go to step 1.

4. If $\theta(w_{opt}^{\delta_k}) < c - \epsilon$, do a binary search with $\delta$ lying between lower limit as $\delta_{\min} = \delta_{k-1}$ and upper limit as $\delta_{\max} = \delta_k$ till we obtain a $\delta$ such that $|\theta(w_{opt}^{\delta}) - c| \leq \epsilon$.

---

In practice, we are interested in obtaining $s$-values over an arbitrary range of constants $c$, in which case, we can just fix a range of values for the penalty $\delta$ in increasing order (say $\delta_0 < \delta_1 < \ldots < \delta_P$ for some $P \in \mathbb{Z}_+$) and use Algorithm 1 to obtain corresponding $s$-value for a given $\delta \in \{\delta_1, \ldots \delta_P\}$ where we can now use warm start to initialize the algorithm for $\delta_p$ using the final iterate of the algorithm for $\delta_{p-1}$. Such heuristics give efficiency gain in practice.

**Remark** The above procedure generalizes to the directional case (2). However, it requires obtaining the conditional expectation of the gradient of the parameter $\theta$ with respect to the variable $E$. We can get an exact estimate of the conditional expectation when $E$ has finite support and similar analysis as above guarantees convergence of the iterates to local optima. However, if $E$ has infinite support (for example, $E$ is a continuous random variable) then we can only obtain an approximation of the conditional expectation using (say) any non-parametric regression method in which case we do not have a guaranteed convergence to local optima. In such situations, we can modify the problem by discretizing $E$ to have such guarantees. We give more details in the next section G.

### F.3 Proof of Proposition 2

We next proceed to prove Proposition 2. The proof uses similar arguments as the proof of Proposition 12.4.4 of Lange [28]. The proof builds on the following lemmas.

**Definition F.3** (Cluster point of a sequence). *A point $w^*$ is a cluster point of a sequence $w^k$ provided there is a subsequence $w^{k_l}$ that tends to $w^*$.*

**Lemma F.1** (Proposition 12.4.1, Lange [28]). *If a bounded sequence $w^k \in \mathbb{R}^n$ satisfies*

$$\lim_{k \to \infty} \|w^{k+1} - w^k\| = 0,$$

*then its set $T$ of cluster points is connected. If $T$ is finite, then $T$ reduces to a single point, and $\lim_{k \to \infty} w^k = w^*$ exists.*

**Lemma F.2.** *Let $\Gamma$ be the set of cluster points generated by the MM sequence $w^{k+1} = M(w^k)$ starting from some initial $w^0$. then $\Gamma$ is contained in the set $S$ of stationary points of (25).*

*Proof.* First observe that the iteration map $M$ in (28) is continuous as $\theta$ is continuously differentiable. Now, the sequence $w^k$ stays within the compact set $W_n$. Consider a cluster point $z = \lim_{l \to \infty} w^{k_l}$. Since the sequence $g(w^k)$ is monotonically decreasing and bounded below, $\lim_{k \to \infty} g(w^k)$ exists. Hence, taking limits in the inequality $g(M(w^k)) \leq g(w^k)$ and using the continuity of functions $M$ and $g$ imply $g(M(z)) = g(z)$. Thus, $z$ is a fixed point of $M$ and also a stationary point of (25). $\square$

**Lemma F.3.** *The set of cluster points $\Gamma$ of $w^{k+1} = M(w^k)$ is compact and connected.*

*Proof.* $\Gamma$ is a closed subset of the compact set $W_n$ and is hence, compact. By Lemma F.1, $\Gamma$ is connected provided $\lim_{k \to \infty} \|w^{k+1} - w^k\| = 0$. If this sufficient condition fails, then by compactness of $W_n$, we can extract a subsequence $w^{k_l}$ such that $\lim_{l \to \infty} w^{k_l} = u$ and $\lim_{l \to \infty} w^{k_l+1} = v$ both

exist, however, $v \neq u$. Further, continuity of function $M$ requires $v = M(u)$ while the descent condition implies

$$g(v) = g(M(u)) = g(u) = \lim_{k \to \infty} g(w^k).$$

Hence, $u$ is a fixed point of $M$, which is a contradiction. Hence, the sufficient condition that $\lim_{k \to \infty} \left\| w^{k+1} - w^k \right\| = 0$ holds. □

From (26), we observe that $G_L(w', w)$ is strictly convex in $w'$ and hence, we have the following chain of inequalities

$$g(w^{k+1}) \leq G_L(w^{k+1}, w^k) < G_L(w^k, w^k) = g(w^k). \tag{33}$$

Since $g$ is lower bounded, hence the sequence $g(w^k)$ decreases and converges which proves 1.

Now, if all stationary points of (25) are isolated and since the domain $W_n$ is compact, then there can only be a finite number of stationary points as an infinite number of them would admit a convergent sequence whose limit will not be isolated. Since, the set of cluster points $\Gamma$ of $w^{k+1} = M(w^k)$ is a connected subset of the finite set of stationary points, $\Gamma$ is a singleton, and hence, the bounded sequence $w^k$ has the single element of $\Gamma$ as its limit. Let $\lim_{k \to \infty} w^k = w^*$, then by Proposition 1, we have $w_i^* \propto e^{-\delta \nabla_i \theta(w^k)}$ for all $i = \{1, 2, \ldots, n\}$. Hence, by Corollary F.1, we have the result.

## G  Directional $s$-values of general estimands

Here, we want to obtain directional $s$-values (with respect to some variable $E$) as in (2) for more general one dimensional parameters defined over the space of probability distributions, $\theta : \mathcal{P} \to \mathbb{R}$. We first characterize the form of a locally optimal solution of the optimization problem in (2) and present algorithm to solve the corresponding optimization problem in Appendix G.1.

We assume that random variable $E$ has finite support of size $K$ (say) and $E$ takes values in the set $\{e_1, \ldots, e_K\}$. We consider a finite sample setting where we observe $n$ samples $\{Z_i, E_i\}_{i=1}^n \overset{\text{i.i.d.}}{\sim} P_0$ for some distribution $P_0 \in \mathcal{P}$ where $\{Z_i, E_i\}_{i=1}^n$ are i.i.d. realizations of the random variable $(Z, E)$. Let the empirical distribution of $\{Z_i, E_i\}_{i=1}^n$ be denoted by $P_{0,n} = \sum_{i=1}^n \frac{1}{n} \delta_i$, where $\delta_i$ is a dirac measure on $(Z_i, E_i)$. We recall that $W_n = [0,1]^n$ is $n$ dimensional unit cube and $S_n = \{w \in \mathbb{R}^n : w_0 + \ldots + w_n = 1, w_i \geq 0 \text{ for } i = 1, \ldots, n\}$ is $n$ dimensional probability simplex. Let $P_w$ denote the probability distribution corresponding to $w \in S_n$ that is, it puts mass $w_i$ on the $i$th sample. We focus on one dimensional parameter $\theta : S_n \to \mathbb{R}$ where we define for $w \in S_n$, $\theta(w)$ as $\theta(\sum_{i=1}^n w_i \delta_i)$. With a slight abuse of notation from now on, we redefine $\theta$ on the $n$ dimensional unit cube $W_n$ as $\theta(w) = \theta\left(\frac{\sum_{i=1}^n w_i \delta_i}{\sum_i w_i}\right)$ for $w \in W_n$. We recall that we want to obtain the conditional $s$-value of parameter $\theta$ (with respect to the variable $E$) given by

$$s_E(\theta - c, P_{0,n}) = \exp\{-\min_{w \in W} \sum_{i=1}^n w_i \log(n w_i)\} \quad \text{s.t. } \theta(w) = c, \ \sum_{i=1}^n w_i = 1 \text{ and}$$
$$P_{0,n}(\cdot \mid E = e_k) = P_w(\cdot \mid E = e_k) \text{ for all } k \in [K]. \tag{34}$$

The constraints $P_{0,n}(\cdot \mid E = e_k) = P_w(\cdot \mid E = e_k)$ for all $k \in [K]$ are linear in weights $w$ that we justify next. Let $I_k$ denote the set of indices such that $E_j = e_k$ for all $j \in I_k$ and each $k \in [K]$, then we have for each $k \in [K]$ and $i \in I_k$

$$P_{0,n}(Z_i \mid E = e_k) = P_w(Z_i \mid E = e_k)$$
$$\implies \frac{w_i}{\sum_{j \in I_k} w_j} = \frac{1}{|I_k|}. \tag{35}$$

Hence, the above constraint implies that for each $k \in [K]$, all $w_i$ such that $i \in I_k$ are equal. That is, the constraints $P_{0,n}(\cdot \mid E = e_k) = P_w(\cdot \mid E = e_k)$ for all $k \in [K]$ are equivalent to the constraint that $w_i = w_j$ for all $(i, j)$ with $E_i = E_j$. We can rewrite the above constraints by a

collection of pairwise equality constraints using a minimum collection of functions $\mathcal{U}$ such that for any $u : W_n \to \mathbb{R}$ such that $u \in \mathcal{U}$, $u$ is given by $u(w) = w_a - w_b$ for some $a \neq b$ where $a, b \in [n]$. Hence, the above optimization problem belongs to the class of general constrained minimization problems with equality constraints (see Chapter 3 of Bertsekas [5]). Now we present necessary and sufficient conditions for a point to be a local optimum of (34), which can be used to verify that we obtained a locally optimal solution of our optimization problem (34). Let $M$ be a random variable taking values in the set $\{\nabla_1\theta(w), \ldots, \nabla_n\theta(w)\}$. Now, for any given probability distribution $P \in \mathcal{P}$, let there be a probability distribution $Q$ such that $\{Z_i, E_i, M_i\}_{i=1}^n \overset{\text{i.i.d.}}{\sim} Q$ where $Q$ is the push-forward of $(Z, E) \sim P$, that is, $Q((Z, E, M) = (Z_i, E_i, \nabla_i\theta)) = P((Z, E) = (Z_i, E_i))$ for $i \in [n]$ and $P \in \mathcal{P}$. In particular, we denote the push forward of $(Z, E) \sim P_0$ under the above mapping by $Q_0$.

We first give a necessary condition for a point to be a local optimum of (34) that follows from Proposition 3.1.1 of Bertsekas [5].

**Corollary G.1** (Necessary conditions). *Assume that $\theta : \text{int}(W_n) \to \mathbb{R}$ is continuously differentiable. Let $w^* \in S_n$ be a locally optimal solution to problem (34), and assume that there does not exist a constant $r \in \mathbb{R}$ such that $(\mathbb{E}_{Q_0}[M \mid E = e_1], \ldots, \mathbb{E}_{Q_0}[M \mid E = e_K]) = r(1, \ldots, 1)$. Then there exists a constant $\lambda \in \mathbb{R}$ such that*

$$w_i^* \propto e^{\lambda \mathbb{E}_{Q_0}[M \mid E = E_i]} \text{ for all } i = \{1, \ldots, n\}. \tag{36}$$

*Proof.* Under the given assumption of Corollary G.1, the assumption that vectors $\nabla\theta$, $(1, \ldots, 1)$, $\nabla_w u$ for $u \in \mathcal{U}$ are linearly independent holds as otherwise we get a contradiction. Now, without loss of generality, we assume that $Z_i$'s are distinct and let $E_1 = E_2 = \ldots = E_m = e_1$ for some $m < n$. We show that

$$w_1 = w_2 = \ldots = w_m \propto e^{\lambda \mathbb{E}_{P_0}[M \mid E = e_1]}.$$

We need to take the derivative of the Lagrangian (38). Without loss of generality, let the functions in $\mathcal{U}$ corresponding to the pair wise equality of $w_1, w_2, \ldots, w_m$ be given by

$$u_1(w) = w_1 - w_2$$
$$u_2(w) = w_1 - w_3$$
$$u_3(w) = w_1 - w_3$$
$$\vdots$$
$$u_{m-1}(w) = w_1 - w_m.$$

Other functions $u \in \mathcal{U}$ do not depend on any of $w_1, \ldots, w_m$.

Hence, the Lagrangian now becomes

$$h(w, \delta, \mu) = \sum_{i=1}^n w_i \log(w_i) + \delta(\theta(w) - c) + \mu(\sum_{i=1}^n w_i - 1) + \sum_{k=1}^{m-1} \alpha_i(w_1 - w_{i+1}) + \sum_{u \in \mathcal{U} - \{u_1, \ldots, u_{m-1}\}} \alpha_u u$$

for $w \in W_n$, and $\delta, \mu, \alpha_u \in \mathbb{R}$.

$$\tag{37}$$

Taking partial derivatives of $h$ with respect to $w_1, \ldots, w_m$, we get

$$\log w_1 + 1 + \delta \nabla_1 \theta(w) + \mu + \alpha_1 + \ldots + \alpha_{m-1} = 0$$
$$\log w_2 + 1 + \delta \nabla_2 \theta(w) + \mu - \alpha_1 = 0$$
$$\vdots$$
$$\log w_m + \delta \nabla_m \theta(w) + \mu - \alpha_{m-1} = 0.$$

Now, invoking the constraint $w_1 = \ldots = w_m$, and adding the above equations, the result follows from Proposition 3.1.1 of Bertsekas [5]. $\qquad\square$

We next present the sufficient condition for a point to be local optima of (34) that again follows from Proposition 3.2.1 of Bertsekas [5]. To that end, we introduce the Lagrangian function $h : \mathbb{R}^n \times \mathbb{R} \times \mathbb{R} \to \mathbb{R}$ that we define as

$$h(w, \delta, \mu) = \sum_{i=1}^{n} w_i \log(w_i) + \delta(\theta(w) - c) + \mu(\sum_{i=1}^{n} w_i - 1) + \sum_{u \in \mathcal{U}} \alpha_u u \text{ for } w \in W_n, \text{ and } \delta, \mu, \alpha_u \in \mathbb{R}. \tag{38}$$

**Corollary G.2** (Second order sufficiency conditions)**.** *Assume that* $\theta : \text{int}(W_n) \to \mathbb{R}$ *is twice continuously differentiable, and let* $w^* \in W_n$, $\delta^*, \mu^* \in \mathbb{R}$ *and* $\alpha^* \in \mathbb{R}^{|U|}$ *satisfy*

$$\nabla_w h(w^*, \delta^*, \mu^*, \alpha^*) = 0, \ \nabla_{\delta,\mu,\alpha} h(w^*, \delta^*, \mu^*, \alpha^*) = 0,$$

$$\gamma' \nabla_{ww}^2 h(w^*, \delta^*, \mu^*, \alpha^*) \gamma > 0, \text{ for all } \gamma \neq 0 \text{ with}$$

$$\nabla \theta(w^*)' \gamma = 0, \sum_{i=1}^{n} \gamma_i = 0 \text{ and } \nabla u(w^*)' \gamma = 0 \text{ for all } u \in \mathcal{U}.$$

*Then* $w^*$ *is a strict local optima of* (34)*.*

Next, we present a Majorization-minimization based algorithm to solve (34) that relies on this characterization.

## G.1 Algorithms to obtain directional $s$-values of general estimands

Here, we solve the optimization problem in (34). Following similar arguments as in Section F.1, we solve the Lagrangian form given by

$$\underset{w_1,\dots,w_n, w_i \geq 0, \sum w_i = 1, P_{0,n}(\cdot|E) = P_w(\cdot|E)}{\text{minimize}} g(w) = \delta(\theta(w) - c) + \sum_{i=1}^{n} w_i \log(w_i). \tag{39}$$

We solve (39) using Majorization-Minimization algorithm. We obtain the majorizer of the objective function in (39) using (24) under Assumption 4 as follows

$$g(w') \leq G_L(w', w) \coloneqq \delta\left(\theta(w) - c + \langle \nabla\theta(w), w' - w \rangle + L \sum_{i=1}^{n} w_i' \log \frac{w_i'}{w_i}\right) + \sum_{i=1}^{n} w_i' \log w_i' \tag{40}$$

for $w, w' \in W_n$.

First we observe that $\langle \nabla\theta(w), w' - w \rangle = \mathbb{E}_{Q_{w'}}[M] - \mathbb{E}_{Q_w}[M]$. Now we want to minimize the right hand side of inequality (39) with respect to $w'$ under the additional constraint $P_{0,n}(\cdot \mid E = e_k) = P_{w'}(\cdot \mid E = e_k)$ for all $k \in [K]$ which gives

$$\langle \nabla\theta(w), w' - w \rangle = \mathbb{E}_{Q_{w'}}[M] - \mathbb{E}_{Q_w}[M]$$
$$= \mathbb{E}_{Q_{w'}}[\mathbb{E}_{Q_{w'}}[M \mid E]] - \mathbb{E}_{Q_w}[M]$$
$$= \mathbb{E}_{Q_{w'}}[\mathbb{E}_{Q_{0,n}}[M \mid E]] - \mathbb{E}_{Q_w}[M].$$

Hence, under the additional constraint, the majorizer now becomes

$$g(w') \leq G_L(w', w) \coloneqq \delta\left(\theta(w) - c + \mathbb{E}_{Q_{w'}}[\mathbb{E}_{Q_{0,n}}[M \mid E]] - \mathbb{E}_{Q_w}[M] + L \sum_{i=1}^{n} w_i' \log \frac{w_i'}{w_i}\right)$$
$$+ \sum_{i=1}^{n} w_i' \log w_i' \tag{41}$$

for $w, w' \in W_n$.

We next show that minimizing the majorizer $G_L(w', w)$ actually involves solving a $K$-dimensional convex optimization problem. The random variable $E$ takes values in the set $\{e_1, \ldots, e_K\}$. Suppose out of the $n$ realizations $\{Z_i, E_i\}_{i=1}^n$, $e_k$ occurs $n_k$ times for $k \in [K]$ and $\sum_{k=1}^K n_k = n$. Now under the constraint $P_{0,n}(\cdot \mid E) = P_{w'}(\cdot \mid E)$, it is equivalent to considering only probability distributions on the set $\{e_1, \ldots, e_K\}$ as conditional on $E = e_k$ for any $k \in [K]$, the corresponding samples are equally likely to occur. Hence, now we can restrict our domain to $K$ dimensional unit cube $W_K$ and minimizing the majorizer in (41) is equivalent to solving the following optimization problem

$$\underset{v' \in W_K, \sum_{k=1}^K v_k' = 1}{\text{minimize}} \ \delta \left( \sum_{k=1}^K v_k' \mathbb{E}_{Q_{0,n}}[M \mid E = e_k] + L \sum_{k=1}^K v_k' \log \frac{v_k'}{v_k} \right) + \sum_{k=1}^K v_k' \log \frac{v_k'}{n_k} \quad (42)$$

which is a $K$ dimensional convex optimization problem. Hence, the convergence analysis follows as in Section F.1.

If the variable $E$ is continuous-valued, then we can discretize $E$ to use the similar procedure as outlined above or use any non-parametric estimator to approximate the conditional expectation $\mathbb{E}_{Q_0}[M \mid E]$.

# H   Confidence intervals

In this section, we prove a general theorem that contains the asymptotic normality results from Section C as a special case. In particular, Lemma C.2 can be recovered with $E = Z = \ell(\eta, Z)$ and $\hat{f}_n(E) = E$, $f(E) = E$. Lemma C.4 can be recovered with $Z = \ell(\eta, Z)$.

**Theorem 3.** *Let $\hat{f}_n(\cdot)$ be an estimate of $f(\cdot) = \mathbb{E}_{P_0}[\ell(\eta, Z) | E = \cdot]$. We assume that $\hat{f}_n$ and $\hat{\lambda}$ are fit on a held-out portion of the data set, that is $\hat{f}_n(\cdot)$ and $\hat{\lambda}$ are independent of $(Z_i, E_i), i = 1, \ldots, n$. We assume that $\sup_{e \in \mathcal{E}} |\hat{f}_n(e) - f(e)| = o_P(n^{-1/4})$. Furthermore, we assume that the moment generating function of $\ell(\eta, Z_i)$ is finite on $\mathbb{R}^p$ and that $\mathbb{E}[f(E)f(E)^\intercal e^{(\lambda^*)^\intercal f(E)}] > 0$. Let $\hat{\lambda} = \arg \min \frac{1}{n} \sum_{i=1}^n e^{\lambda^\intercal \hat{f}_n(E_i)}$ and $\lambda^* = \arg \min \mathbb{E}_{P_0}[e^{(\lambda)^\intercal f(E)}]$. Then,*

$$\frac{1}{n} \sum_{i=1}^n (1 + \hat{\lambda}^\intercal \ell(\eta, Z_i) - \hat{\lambda}^\intercal \hat{f}_n(E_i)) e^{\hat{\lambda}^\intercal \hat{f}_n(E_i)} - \mathbb{E}_{P_0}[e^{(\lambda^*)^\intercal f(E)}] \overset{d}{=} \mathcal{N}\left(0, \frac{\sigma^2}{n}\right) + o_P(1/n),$$

*where*

$$\sigma^2 = Var_{P_0}(e^{\lambda^* f(E)}) + Var_{P_0}(e^{(\lambda^*)^\intercal f(E)}(\lambda^*)^\intercal (\ell(\eta, Z) - f(E))).$$

*Proof.* Using Lemma C.3, we have $\hat{\lambda} \to \lambda^*$ in probability. By definition of $\hat{\lambda}$,

$$\frac{1}{n} \sum_{i=1}^n \hat{f}_n(E_i) e^{\hat{\lambda}^\intercal \hat{f}_n(E_i)} = 0.$$

Using a Taylor expansion on the left,

$$\frac{1}{n} \sum_{i=1}^n f(E_i) e^{\hat{\lambda}^\intercal f(E_i)} = o_P(n^{-1/4}).$$

Thus, using another Taylor expansion,

$$(\hat{\lambda} - \lambda^*)^\intercal \frac{1}{n} \sum_{i=1}^n f(E_i) f(E_i)^\intercal e^{(\lambda^*)^\intercal f(E_i)} = O_P(\|\hat{\lambda} - \lambda^*\|_2^2 + \frac{1}{\sqrt{n}}) + o_P(n^{-1/4}).$$

Since $\hat{\lambda} - \lambda^* \to 0$ and $\mathbb{E}[f(E)f(E)^\intercal e^{(\lambda^*)^\intercal f(E)}] > 0$ we have

$$\hat{\lambda} - \lambda^* = o_P(n^{-1/4}).$$

To summarize, we know that $\hat{\lambda} - \lambda^* = o_P(n^{-1/4})$ and that $\sup_{e \in \mathcal{E}} |\hat{f}_n(e) - f(e)| = o_P(n^{-1/4})$. Thus,

$$\frac{1}{n}\sum_{i=1}^{n} e^{\hat{\lambda}^\intercal \hat{f}_n(E_i)} - \mathbb{E}[e^{(\lambda^*)^\intercal f(E)}] = (\hat{\lambda} - \lambda^0)^\intercal \frac{1}{n}\sum_{i=1}^{n} f(E_i) e^{(\lambda^*)^\intercal f(E_i)}$$

$$+ \frac{1}{n}\sum_{i=1}^{n}(\hat{\lambda})^\intercal(\hat{f}_n(E_i) - f(E_i)) e^{(\hat{\lambda})^\intercal \hat{f}_n(E_i)}$$

$$+ \frac{1}{n}\sum_{i=1}^{n} e^{(\lambda^*)^\intercal f(E_i)} - \mathbb{E}_{P_0}[e^{(\lambda^*)^\intercal f(E)}] + o_P(n^{-1/2})$$

Using the CLT, the first term goes to zero at rate $O_P(n^{-3/4})$. The last term can be computed with a CLT (since we assume that the moment generating function is finite, the variance is finite). Let us focus on the second term. Note that the second term in the previous equation can be re-written:

$$\frac{1}{n}\sum_{i=1}^{n}(\hat{\lambda})^\intercal(\hat{f}_n(E_i) - f(E_i)) e^{(\hat{\lambda})^\intercal \hat{f}_n(E_i)} = \frac{1}{n}\sum_{i=1}^{n}(\hat{\lambda})^\intercal(\ell(\eta, Z_i) - f(E_i)) e^{(\hat{\lambda})^\intercal \hat{f}_n(E_i)}$$

$$+ \frac{1}{n}\sum_{i=1}^{n}(\hat{\lambda})^\intercal(\hat{f}_n(E_i) - \ell(\eta, Z_i)) e^{(\hat{\lambda})^\intercal \hat{f}_n(E_i)}$$

Using the last two equations,

$$\frac{1}{n}\sum_{i=1}^{n}(1 + \hat{\lambda}^\intercal \ell(\eta, Z_i) - \hat{\lambda}^\intercal \hat{f}_n(E_i)) e^{\hat{\lambda}^\intercal \hat{f}_n(E_i)} - \mathbb{E}_{P_0}[e^{(\lambda^*)^\intercal f(E)}]$$

$$= \frac{1}{n}\sum_{i=1}^{n}(\hat{\lambda})^\intercal(\ell(\eta, Z_i) - f(E_i)) e^{(\hat{\lambda})^\intercal \hat{f}_n(E_i)} + \frac{1}{n}\sum_{i=1}^{n} e^{(\lambda^*)^\intercal f(E_i)} - \mathbb{E}_{P_0}[e^{(\lambda^*)^\intercal f(E)}] + o_P(n^{-1/2})$$

(43)

Using that

$$\mathbb{E}_{P_0}[(\ell(\eta, Z) - f(E))|E] = 0,$$

by conditioning on the $E_1, \ldots, E_n$ using a CLT we get

$$\frac{1}{n}\sum_{i=1}^{n}(\ell(\eta, Z_i) - f(E_i)) e^{(\hat{\lambda})^\intercal \hat{f}_n(E_i)} = \frac{1}{n}\sum_{i=1}^{n}(\ell(\eta, Z_i) - f(E_i)) e^{(\lambda^*)^\intercal f(E_i)} + o_P(1/\sqrt{n}).$$

Here we used that $\hat{f}_n$ is computed on a separate data set and thus is independent of $(Z_i, E_i)$, $i = 1, \ldots, n$. Furthermore, we used that $\hat{\lambda}$ depends on the $(Z_i, E_i)$ only through the $E_i$. Using this in equation (43), we get

$$\frac{1}{n}\sum_{i=1}^{n}(1 + \hat{\lambda}^\intercal \ell(\eta, Z_i) - \hat{\lambda}^\intercal \hat{f}_n(E_i)) e^{\hat{\lambda}^\intercal \hat{f}_n(E_i)} - \mathbb{E}_{P_0}[e^{(\lambda^*)^\intercal f(E)}]$$

$$= \frac{1}{n}\sum_{i=1}^{n}(\lambda^*)^\intercal(\ell(\eta, Z_i) - f(E_i)) e^{(\lambda^*)^\intercal f(E_i)}$$

$$+ \frac{1}{n}\sum_{i=1}^{n} e^{(\lambda^*)^\intercal f(E_i)} - \mathbb{E}_{P_0}[e^{(\lambda^*)^\intercal f(E)}] + o_P(1/\sqrt{n})$$

Thus, asymptotically, we have that

$$\frac{1}{n}\sum_{i=1}^{n}(1 + \hat{\lambda}^\intercal \ell(\eta, Z_i) - \hat{\lambda}^\intercal \hat{f}_n(E_i)) e^{\hat{\lambda}^\intercal \hat{f}_n(E_i)} - \mathbb{E}_{P_0}[e^{(\lambda^*)^\intercal f(E)}] \overset{d}{=} \mathcal{N}(0, \sigma^2) + o_P(1/\sqrt{n}),$$

where

$$\sigma^2 = \frac{1}{n}\text{Var}(e^{(\lambda^*)^\intercal f(E_i)}) + \frac{1}{n}\text{Var}(e^{(\lambda^*)^\intercal f(E_i)}(\lambda^*)^\intercal(\ell(\eta, Z) - f(E))).$$

$\square$

# I    Other technical proofs and appendices

**Theorem 4.** *[Theorem II.1, Andersen and Gill [1]] Let $E$ be an open convex subset of $\mathbb{R}^p$ and let $F_1, F_2, \ldots,$ be a sequence of random concave functions on $E$ such that $F_n(x) \xrightarrow{P} f(x)$ as $n \to \infty$ for every $x \in E$, where $f$ is some real function on $E$. Then $f$ is also concave and for all compact $A \subset E$,*

$$\sup_{x \in A} |F_n(x) - f(x)| \xrightarrow{P} 0 \text{ as } n \to \infty.$$

**Corollary I.1.** *[Corollary II.1, Andersen and Gill [1]] Let $E$ be an open convex subset of $\mathbb{R}^p$ and let $F_1, F_2, \ldots,$ be a sequence of random concave functions on $E$ such that $F_n(x) \xrightarrow{P} f(x)$ as $n \to \infty$ for every $x \in E$, where $f$ is some real function on $E$. Suppose $f$ has a unique maximum at $\hat{x} \in E$. Let $\hat{X}_n$ maximize $F_n$. Then $\hat{X}_n \xrightarrow{P} \hat{x}$ as $n \to \infty$.*

**Corollary I.2.** *Let $E$ be an open convex subset of $\mathbb{R}^p$ and let $F_1, F_2, \ldots,$ be a sequence of random concave functions on $E$ such that $F_n(x) \xrightarrow{P} f(x)$ as $n \to \infty$ for every $x \in E$, where $f$ is some real function on $E$. Suppose $f$ has a unique maximum at $\hat{x} \in E$. Let $\hat{X}_n$ maximize $F_n$. Then $F_n(\hat{X}_n) \xrightarrow{P} f(\hat{x})$ as $n \to \infty$.*

*Proof.* We define a set $B$ as $B = \{x : \|x - \hat{x}\| \leq \gamma\}$ for some arbitrary small $\gamma > 0$ such that $B \subseteq E$. Clearly, set $B$ is compact. From Corollary I.1, we have $\hat{X}_n \xrightarrow{P} \hat{x}$. Hence, there exists positive integer $N_1$ such that $\hat{X}_n \in B$ for all $n > N_1$ with probability at least $1 - \delta$ for some small $\delta > 0$.

Since $\sup_{x \in B} |F_n(x) - f(x)| \xrightarrow{P} 0$. Hence, for any $\epsilon > 0$, there exists positive integer $N_2$ such that

$$|F_n(x) - f(x)| < \epsilon \text{ for all } x \in B \text{ and } n > N_2 \tag{44}$$

with probability at least $1 - \delta$. Let $x_0 \in B$ be such that $f(x_0) \geq \sup_{x \in B} f(x) - \epsilon$. Hence, using (44), we have for all $n > N_2$

$$\sup_{x \in B} f(x) \leq f(x_0) + \epsilon \leq F_n(x_0) + 2\epsilon \leq \sup_{x \in B} F_n(x) + 2\epsilon \tag{45}$$

with probability at least $1 - \delta$.

Now, we choose sequence $x_n \in B$ such that $F_n(x_n) \geq \sup_{x \in B} F_n(x) - \epsilon$. Using (44), we have for all $n > N_2$

$$\sup_{x \in B} f(x) + \epsilon \geq F_n(x_n) \geq \sup_{x \in B} F_n(x) - \epsilon \tag{46}$$

with probability at least $1 - \delta$. Combining (45) and (46), we have for all $n > N_2$,

$$|\sup_{x \in B} F_n(x) - \sup_{x \in B} f(x)| < 2\epsilon \tag{47}$$

with probability at least $1 - \delta$. We choose $N = \max\{N_1, N_2\}$. Since, $\hat{X}_n \in B$ for all $n > N$ with probability at least $1 - \delta$. We have for all $n > N$, with probability at least $1 - 2\delta$,

$$|F_n(\hat{X}_n) - f(\hat{x})| < 2\epsilon. \tag{48}$$

Hence, the proof follows. $\qquad\square$

## I.1    Proof of Lemma C.1

First, if $Z \geq 0$ with probability 1 or if $Z \leq 0$ with probability 1, the statement is immediate. Thus, in the following we assume that $Z > 0$ with non-vanishing probability and that $Z < 0$ with non-vanishing probability.

Let us now prove that the minimum is achieved for some unique $\lambda^* \in \mathbb{R} \cup \{-\infty, \infty\}$. We will do the proof by contradiction. If the minimum is attained for multiple $\lambda^*$, then by convexity there must exist a nonempty open interval $(\lambda_1, \lambda_2)$ of values $\lambda^*$ that attain the minimum. Using a second order Taylor

expansion, one can show that in this case we must have $Z \equiv c$ almost surely. However, we assumed that $Z$ is non-degenerate. Thus, the minimum is achieved for some unique $\lambda^* \in \mathbb{R} \cup \{-\infty, \infty\}$.

Furthermore, if $Z > 0$ with probability $> 0$ then $\mathbb{E}[e^{\lambda Z}] \to \infty$ for $\lambda \to \infty$. Similarly if $Z < 0$ with probability $< 0$ then $\mathbb{E}[e^{\lambda Z}] \to \infty$ for $\lambda \to -\infty$. Thus, the minimum is achieved for $\lambda^* \in \mathbb{R}$.

The proof follows from Corollary I.2 and using the fact that the negative of a convex function is concave.

## I.2 Proof of Theorem 2

*Proof.* Any distribution $\mathbb{P}$ that satisfies $\mathbb{P}[\cdot|E = e] = \mathbb{P}_0[\cdot|E = e]$ for all $e \in \mathcal{E}$ satisfies

$$\mathbb{E}_P[Z] = \mathbb{E}_P[\mathbb{E}_{P_0}[Z|E]]. \tag{49}$$

Thus,

$$s_E(\theta, P_0) = \exp\{- \min_{P \in \mathcal{P}: P(\cdot|E=e)=P_0(\cdot|E=e) \text{ for all } e \in \mathcal{E}} D_{KL}(P||P_0)\} \quad \text{s.t.} \quad \mathbb{E}_P[Z] = 0.$$
$$= \exp\{- \min_{P \in \mathcal{P}: P(\cdot|E=e)=P_0(\cdot|E=e) \text{ for all } e \in \mathcal{E}} D_{KL}(P||P_0)\} \quad \text{s.t.} \quad \mathbb{E}_P[\mathbb{E}_{P_0}[Z|E]] = 0.$$

Since $\mathbb{E}_{P_0}[Z|E]$ is a function of $E$, using the chain rule for KL divergence,

$$s_E(\theta, P_0) = \exp\{- \min_{P \in \mathcal{P}: P(\cdot|E=e)=P_0(\cdot|E=e) \text{ for all } e \in \mathcal{E}} D_{KL}(P||P_0)\} \quad \text{s.t.} \quad \mathbb{E}_P[\mathbb{E}_{P_0}[Z|E]] = 0$$
$$= \exp\{- \min_{P \in \mathcal{P}} D_{KL}(P||P_0)\} \quad \text{s.t.} \quad \mathbb{E}_P[\mathbb{E}_{P_0}[Z|E]] = 0.$$

Now we can use Theorem 1 for the random variable $\mathbb{E}_{P_0}[Z|E]$, which completes the proof. $\square$

## I.3 Proof of Lemma C.3

We will show that for any compact subset $\Lambda \subset \mathbb{R}$,

$$\sup_{\lambda \in \Lambda} |E_{P_n}[e^{\lambda \hat{f}_n(E)}] - E_{P_0}[e^{\lambda \mathbb{E}_{P_0}[Z|E]}]| \xrightarrow{P} 0. \tag{50}$$

Since $E_{P_n}[e^{\lambda \hat{f}_n(E)}]$ and $E_{P_0}[e^{\lambda \mathbb{E}_{P_0}[Z|E]}]$ are convex functions in $\lambda$, hence, the proof follows from Corollary I.2. In order to show (50), it suffices to show the following:

$$\sup_{\lambda \in \Lambda} |E_{P_n}[e^{\lambda \hat{f}_n(E)}] - E_{P_n}[e^{\lambda \mathbb{E}_{P_0}[Z|E]}]| \xrightarrow{P} 0 \text{ and} \tag{51}$$

$$\sup_{\lambda \in \Lambda} |E_{P_n}[e^{\lambda \mathbb{E}_{P_0}[Z|E]}] - E_{P_0}[e^{\lambda \mathbb{E}_{P_0}[Z|E]}]| \xrightarrow{P} 0. \tag{52}$$

(52) follows from Theorem 4. We next show (51). Since $Z$ has finite moment generating function, hence, the random variable $\mathbb{E}_{P_0}[Z \mid E]$ also has finite moment generating function. Hence, for any small $\epsilon > 0$, we can choose a large $M \in \mathbb{R}$ such that the set $R = \{e \in \mathbb{R}^d \mid |\mathbb{E}_{P_0}[Z \mid E = e]| \leq M\}$ satisfies $P_0(R) \geq 1 - \epsilon$. Now, from Assumption 1, we have

$$\sup_{\lambda \in \Lambda} \sup_{e \in R} |e^{\lambda \hat{f}_n(e)} - e^{\lambda \mathbb{E}_{P_0}[Z|E=e]}| \xrightarrow{P} 0. \tag{53}$$

Hence,

$$\sup_{\lambda \in \Lambda} |E_{P_n}[e^{\lambda \hat{f}_n(E)} 1\{E \in R\}] - E_{P_n}[e^{\lambda \mathbb{E}_{P_0}[Z|E]} 1\{E \in R\}]| \leq \sup_{\lambda \in \Lambda} \sup_{e \in R} |e^{\lambda \hat{f}_n(e)} - e^{\lambda \mathbb{E}_{P_0}[Z|E=e]}| \xrightarrow{P} 0.$$
$$\tag{54}$$

Since we can choose the set $R$ with an arbitrarily large probability, we have (51).

### I.4 Proof of Corollary D.1

Since $L$ is convex and smooth in its first argument, hence, the minimizer in (10) is equivalently a solution of $E_P[\ell(\theta^M, Z)] = 0$. To obtain $s$-value in (11), we need to find the distribution $P$ closest to $P_0$ such that $E_P[\ell(\eta, Z)] = 0$. Hence, $s$-value in (11) can be rewritten as

$$s(\theta^M - \eta, P_0) = \exp\{-\min_{P \in \mathcal{P}} D_{KL}(P||P_0)\} \quad \text{s.t.} \quad E_P[\ell(\eta, Z)] = 0. \tag{55}$$

This is the same problem as obtaining $s$-value for a multivariate mean of the random variable $\ell(\eta, Z)$. Hence, following similar arguments as the proof for Theorem 1, we have the result.

### I.5 Proof of Corollary D.2

Since $L$ is convex and smooth in its first argument, hence, the minimizer in (10) is equivalently a solution of $E_P[\ell(\theta^M, Z)] = 0$. To obtain $s$-value in (13), we need to find the distribution $P$ closest to $P_0$ such that $P[\bullet \mid E = e] = P_0[\bullet \mid E = e]$ for all $e \in \mathcal{E}$ and $E_P[\ell(\eta, Z)] = 0$. Hence, the $s$-value in (13) can be rewritten as

$$s(\theta^M - \eta, P_0) = \exp\{-\min_{P \in \mathcal{P}, P[\bullet|E=e]=P_0[\bullet|E=e] \text{ for all } e \in \mathcal{E}} D_{KL}(P||P_0)\} \text{ s.t. } E_P[\ell(\eta, Z)] = 0. \tag{56}$$

This is the same problem as obtaining directional $s$-value for the multivariate mean of the random variable $\ell(\eta, Z)$. Hence, following similar arguments as the proof for Theorem 2, we have the result.

**Lemma I.1.** *Let $\mathcal{X} \subseteq \mathbb{R}^m$ be an open convex set and $\mathcal{Y} \subset \mathbb{R}^d$ be a compact set. Let $\{f_n\}_{n \geq 1}$ be a sequence of real valued functions defined on $\mathcal{X} \times \mathcal{Y}$, where each of the function $f_n$ is convex in the first variable and converges pointwise on $\mathcal{X} \times \mathcal{Y}$ to a function $f$, that is*

$$f(x, y) = \lim_{n \to \infty} f_n(x, y) \text{ for all } (x, y) \in \mathcal{X} \times \mathcal{Y}.$$

*Suppose that*

$$g_n(x) = \sup_{y \in \mathcal{Y}} |f_n(x, y) - f(x, y)| \to 0 \text{ for each } x \in \mathcal{X} \text{ as } n \to \infty \tag{57}$$

*and*

$$\sup_{y \in \mathcal{Y}} |f(x, y)| < \infty \text{ for each } x \in \mathcal{X}. \tag{58}$$

*Then $\sup_{y \in \mathcal{Y}} |f_n(x, y) - f(x, y)| \to 0$ uniformly on each compact $S \subset \mathcal{X}$ as $n \to \infty$.*

*Proof.* The proof works along similar lines as the proof of Theorem 10.8 in Rockafellar [37]. First, we observe that the collection $\{f_n(\cdot, y) \mid n \geq 1 \text{ and } y \in \mathcal{Y}\}$ is pointwise bounded on $\mathcal{X}$ using (57) and (58). Hence, by Theorem 10.6 of Rockafellar [37] it is equi-Lipschitzian on each closed bounded subset of $\mathcal{X}$. Then there exists a real number $\alpha > 0$ such that

$$|f_n(x_1, y) - f_n(x_2, y)| \leq \alpha |x_1 - x_2|, \text{ for all } x_1, x_2 \in S, n \geq 1 \text{ and } y \in \mathcal{Y}. \tag{59}$$

Since $S$ is compact, hence, there exists a finite subset $C_0$ of $S$ such that each point of $S$ lies within $\frac{\epsilon}{3\alpha}$ distance of at least one point of $C_0$. Since $C_0$ is finite and the functions $g_n$ converge pointwise on $C_0$, there exists an integer $N_0$ such that

$$|f_{n_1}(x, y) - f_{n_2}(x, y)| \leq \frac{\epsilon}{3\alpha} \text{ for all } n_1, n_2 \geq N_0, x \in C_0 \text{ and } y \in \mathcal{Y}. \tag{60}$$

Given any $x \in S$, let $z$ be one of the points of $C_0$ such that $|z - x| \leq \frac{\epsilon}{3\alpha}$. Then for all $n_1, n_2 \geq N_0$ and $y \in \mathcal{Y}$, we have

$$|f_{n_1}(x, y) - f_{n_2}(x, y)| \leq |f_{n_1}(x, y) - f_{n_1}(z, y)| + |f_{n_1}(z, y) - f_{n_2}(z, y)| + |f_{n_2}(z, y) - f_{n_2}(x, y)|$$

$$\leq \alpha|x - z| + \frac{\epsilon}{3} + \alpha|z - x| \leq \epsilon.$$

Hence, the sequence $\{f_n\}_{n \geq 1}$ is cauchy uniformly in $x \in S$ and $y \in \mathcal{Y}$. Hence, the proof follows. $\quad\square$

**Lemma I.2.** *Let $\mathcal{X} \subseteq \mathbb{R}^m$ be an open convex set and $\mathcal{Y} \subset \mathbb{R}^d$ be a compact set. Let $\{F_n\}_{n \geq 1}$ be a sequence of real valued random functions defined on $\mathcal{X} \times \mathcal{Y}$, where each of the function $F_n$ is convex in the first variable.*

*Suppose that*

$$g_n(x) = \sup_{y \in \mathcal{Y}} |F_n(x, y) - f(x, y)| \xrightarrow{P} 0 \text{ for each } x \in \mathcal{X} \text{ as } n \to \infty \text{ and} \tag{61}$$

$$\sup_{y \in \mathcal{Y}} |f(x, y)| < \infty \text{ for each } x \in \mathcal{X}. \tag{62}$$

*Then $\sup_{y \in \mathcal{Y}} |F_n(x, y) - f(x, y)| \xrightarrow{P} 0$ uniformly on each compact $S \subset \mathcal{X}$ as $n \to \infty$.*

*Proof.* The proof uses subsequence arguments very similar to that in the proof of Theorem II.1 of Andersen and Gill [1]. Let $x_1, x_2, \ldots$ be a countable dense set of points in $\mathcal{X}$. Since $g_n(x_1) \xrightarrow{P} 0$ as $n \to \infty$ there exists a subsequence along which convergence holds almost surely. Along this subsequence $g_n(x_2) \xrightarrow{P} 0$, hence, a further subsequence exists along which $g_n(x_2) \xrightarrow{\text{a.s.}} 0$. By repeating the argument, along a $\text{sub}_k$ sequence, $g_n(x_j) \xrightarrow{\text{a.s.}} 0$ for $j = 1, \ldots, k$. By considering the new subsequence formed by taking the first element of the first subsequence, the second element of the second subsequence and so on, we have $g_n(x_j) \xrightarrow{\text{a.s.}} 0$ for each $j = 1, 2, \ldots$.

Hence, by Lemma I.1, it follows that

$$\sup_{x \in S} g_n(x) \xrightarrow{\text{a.s.}} 0 \text{ along this subsequence.}$$

Since, for any subsequence, there exists a further subsequence along which $\sup_{x \in S} g_n(x) \xrightarrow{\text{a.s.}} 0$. It then follows that $\sup_{x \in S} g_n(x) \xrightarrow{P} 0$ along the whole sequence. $\qquad\square$

## I.6 Proof of Lemma D.1

By Assumption 2, it follows that $\sup_{\eta \in \Sigma} |E_{P_n}[e^{\lambda^\top \ell(\eta, Z)}] - E_{P_0}[e^{\lambda^\top \ell(\eta, Z)}]| \xrightarrow{P} 0$ (see Theorem 19.4 and Example 19.8 of van der Vaart [47]). Let $\Lambda \subset \mathbb{R}^p$ be a compact subset. Since $e^{\lambda^\top \ell(\eta, Z)}$ is convex in $\lambda$, by Assumption 2 and Lemma I.2, we have $\sup_{\lambda \in \Lambda} \sup_{\eta \in \Sigma} |E_{P_n}[e^{\lambda^\top \ell(\eta, Z)}] - E_{P_0}[e^{\lambda^\top \ell(\eta, Z)}]| \xrightarrow{P} 0$.

Let $f_n(\lambda, \eta) = E_{P_n}[e^{\lambda^\top \ell(\eta, Z)}]$ and $f(\lambda, \eta) = E_{P_0}[e^{\lambda^\top \ell(\eta, Z)}]$. Since $\sup_\eta \sup_\lambda |f_n(\lambda, \eta) - f(\lambda, \eta)| \xrightarrow{P} 0$, for any $\epsilon > 0$, there exists $N$ such that

$$|f_n(\lambda, \eta) - f(\lambda, \eta)| < \epsilon \text{ for all } \lambda \in \Lambda, \eta \in \Sigma \text{ and } n > N \tag{63}$$

with probability at least $1 - \delta$ for some small $\delta > 0$. We first show that $\sup_{\eta \in \Sigma} |\inf_{\lambda \in \Lambda} f_n(\lambda, \eta) - \inf_\lambda f(\lambda, \eta)| \xrightarrow{P} 0$.

For $\eta \in \Sigma$, let $\lambda_0 \in \Lambda$ be such that $f(\lambda_0, \eta) \leq \inf_\lambda f(\lambda, \eta) + \epsilon$. Hence, using (63), we have for all $n > N$

$$\inf_{\lambda \in \Lambda} f(\lambda, \eta) \geq f(\lambda_0, \eta) - \epsilon \geq f_n(\lambda_0, \eta) - 2\epsilon \geq \inf_\lambda f_n(\lambda, \eta) - 2\epsilon \tag{64}$$

with probability at least $1 - \delta$. Now, for $\eta \in \Sigma$, we choose $\lambda_n \in \Lambda$ such that $f_n(\lambda_n) \leq \inf_\lambda f_n(\lambda, \eta) + \epsilon$. Using (63), we have

$$\inf_{\lambda \in \Lambda} f(\lambda, \eta) - \epsilon \leq f_n(\lambda_n, \eta) \leq \inf_{\lambda \in \Lambda} f_n(\lambda, \eta) + \epsilon \tag{65}$$

with probability at least $1 - \delta$.

Combining (64) and (65), we have

$$|\inf_{\lambda \in \Lambda} f_n(\lambda, \eta) - \inf_{\lambda \in \Lambda} f(\lambda, \eta)| < 2\epsilon \tag{66}$$

for all $\eta$ and $n > N$ with probability at least $1 - \delta$.

Let $g_n(\eta) = \inf_{\lambda \in \Lambda} f_n(\lambda, \eta)$ and $g(\eta) = \inf_{\lambda \in \Lambda} f(\lambda, \eta)$, then $\sup_\eta |g_n(\eta) - g(\eta)| \xrightarrow{P} 0$. Now we need to show $\sup_{\eta_k} |\sup_{\eta_1, \ldots, \eta_{k-1}, \eta_k, \ldots \eta_p} g_n(\eta) - \sup_{\eta_1, \ldots, \eta_{k-1}, \eta_k, \ldots \eta_p} g(\eta)| \xrightarrow{P} 0$, which follows similarly as the proof of (66).

## I.7  Proof of Lemma D.2

We need to show that for any compact subset $\Lambda \subset \mathbb{R}^p$,

$$\sup_{\lambda \in \Lambda} \sup_{\eta \in \Sigma} |E_{P_n}[e^{\lambda^\intercal Q_n(\eta, E)}] - E_{P_0}[e^{\lambda^\intercal \mathbb{E}_{P_0}[\ell(\eta, Z)|E]}]| \xrightarrow{P} 0 \tag{67}$$

and then the rest of the proof follows similarly as in the proof of Lemma D.1. In order to show (67), it suffices to show the following:

$$\sup_{\lambda \in \Lambda} \sup_{\eta \in \Sigma} |E_{P_n}[e^{\lambda^\intercal Q_n(\eta, E)}] - E_{P_n}[e^{\lambda^\intercal \mathbb{E}_{P_0}[\ell(\eta, Z)|E]}]| \xrightarrow{P} 0. \tag{68}$$

$$\sup_{\lambda \in \Lambda} \sup_{\eta \in \Sigma} |E_{P_n}[e^{\lambda^\intercal \mathbb{E}_{P_0}[\ell(\eta, Z)|E]}] - E_{P_0}[e^{\lambda^\intercal \mathbb{E}_{P_0}[\ell(\eta, Z)|E]}]| \xrightarrow{P} 0. \tag{69}$$

(69) follows similarly as in the proof of Lemma D.1. hence, it remains to show show (68).

Since $\Lambda$ is a compact set, there exists a real constant $M$ such that $\|\lambda\|_1 \leq M$ for all $\lambda \in \Lambda$. Hence,

$$|\lambda^\intercal Q_n(\eta, e) - \lambda^\intercal \mathbb{E}_{P_0}[\ell(\eta, Z) \mid E = e]| \leq \|\lambda\|_1 \|Q_n(\eta, e) - \mathbb{E}_{P_0}[\ell(\eta, Z) \mid E = e]\|_\infty$$
$$\leq M \|Q_n(\eta, e) - \mathbb{E}_{P_0}[\ell(\eta, Z) \mid E = e]\|_\infty. \tag{70}$$

Now, for any fixed value of $E = e$, for some $\psi_{n,\lambda,e} \in \mathbb{R}$ such that $\lambda^\intercal Q_n(\eta, e) \leq \psi_{n,\lambda,e} \leq \lambda^\intercal \mathbb{E}_{P_0}[\ell(\eta, Z) \mid E = e]$, we have by Taylor's expansion

$$|e^{\lambda^\intercal Q_n(\eta, e)} - e^{\lambda^\intercal \mathbb{E}_{P_0}[\ell(\eta, Z)|E=e]}| \leq e^{\psi_{n,\lambda,e}} |\lambda^\intercal Q_n(\eta, e) - \lambda^\intercal \mathbb{E}_{P_0}[\ell(\eta, Z) \mid E = e]|$$

$$\leq e^{\lambda^\intercal \mathbb{E}_{P_0}[\ell(\eta, Z)|E=e]} e^{M\|Q_n(\eta, e) - \mathbb{E}_{P_0}[\ell(\eta, Z)|E=e]\|_\infty} M \|Q_n(\eta, e) - \mathbb{E}_{P_0}[\ell(\eta, Z) \mid E = e]\|_\infty \tag{71}$$

where the last inequality follows from (70).

Since by Assumption 3, we have $\sup_\eta \sup_e \|\mathbb{E}_{P_0}[\ell(\eta, Z)|E = e] - Q_n(\eta, e)\|_\infty \to 0$. Hence, in order to show (68), it suffices to show that $\sup_{\lambda \in \Lambda} \sup_{\eta \in \Sigma} \mathbb{E}_{P_n}[e^{\lambda^\intercal \mathbb{E}_{P_0}[\ell(\eta, Z)|E]}] \leq C < \infty$ for some numerical constant $C$ independent of $n$ with high probability. Now, by Assumption 2, we have $\mathbb{E}_{P_0}[\sup_{\eta \in \Sigma} e^{\lambda^\intercal \ell(\eta, Z)}] < \infty$ for any $\lambda \in \mathbb{R}^p$. Hence, by Jensen's inequality, we have $\mathbb{E}_{P_0}[\sup_{\eta \in \Sigma} e^{\lambda^\intercal \mathbb{E}_{P_0}[\ell(\eta, Z)|E]}] < \infty$ for any $\lambda \in \mathbb{R}^p$. Also, $\lambda \to \sup_{\eta \in \Sigma} e^{\lambda^\intercal \mathbb{E}_{P_0}[\ell(\eta, Z)|E=e]}$ is a convex function for any $e$. Hence, by Theorem 4, we have the result.

## J  Additional experimental details

Table 5: National supported work demonstration (NSW) data. Table showing the sample means of covariates for the subset extracted by Dehejia and Wahba [12] (DJW) and the subset containing the remaining samples (DJWC) along with $p$-values for testing the difference of means between the two treated and control groups respectively using Welch two sample t-test.

Age=age in years; Education=number of years of schooling; Black=1 if black, 0 otherwise; Hispanic=1 if Hispanic, 0 otherwise; Nodegree=1 if no high school degree, 0 otherwise; Married =1 if married, 0 otherwise; RE75= Earnings in 1975.

| Part | | | | | |
| --- | --- | --- | --- | --- | --- |
| | No. of Obs. | Age | Education | Black | Hispanic |
| Treated (DJW) | 185 | 25.81 | 10.35 | 0.84 | 0.06 |
| Treated (DJWC) | 112 | 22.66 | 10.44 | 0.73 | 0.15 |
| p-values for diff. in means | | **1.95e-05** | 0.6501 | **0.027** | **0.017** |
| Control (DJW) | 260 | 25.05 | 10.09 | 0.83 | 0.11 |
| Control (DJWC) | 165 | 23.49 | 10.35 | 0.76 | 0.12 |
| p-values for diff. in means | | **0.0123** | 0.1111 | **0.09** | 0.6724 |

Table 6: National supported work demonstration (NSW) data. Table showing the sample means of covariates for the subset extracted by Dehejia and Wahba [12] (DJW) and the subset containing the remaining samples (DJWC) along with $p$-values for testing the difference of means between the two treated and control groups respectively using Welch two sample t-test.

Age=age in years; Education=number of years of schooling; Black=1 if black, 0 otherwise; Hispanic=1 if Hispanic, 0 otherwise; Nodegree=1 if no high school degree, 0 otherwise; Married =1 if married, 0 otherwise; RE75= Earnings in 1975.

| Part | | | | |
| --- | --- | --- | --- | --- |
| | No. of Obs. | Nodegree | Married | RE75 |
| Treated (DJW) | 185 | 0.19 | 0.71 | 1532.1 |
| Treated (DJWC) | 112 | 0.13 | 0.77 | 5600 |
| p-values for diff. in means | | 0.2035 | 0.2539 | **5.42e-10** |
| Control (DJW) | 260 | 0.15 | 0.83 | 1266.9 |
| Control (DJWC) | 165 | 0.16 | 0.78 | 5799.66 |
| p-values for diff. in means | | 0.7891 | 0.1841 | **6.967e-15** |