# OpenReview forum: "The s-value: evaluating stability with respect to distributional shifts"
_NeurIPS.cc/2023/Conference — NeurIPS 2023 poster_

### Official Review · Reviewer_buuF · 2023-06-27

**Soundness:** 2 fair
**Presentation:** 1 poor
**Contribution:** 2 fair
**Rating:** 3
**Confidence:** 2

**Summary:**

The paper presents a metric (s-value) that quantifies the uncertainty of statistical estimators in terms of their distributional instability. In addition, the techniques proposed can quantify the effect of directional shifts and the authors also discuss how the s-value can be used to improve estimation accuracy under shifted distributions.

**Strengths:**

The development of methods that quantify the effect of distributions shifts is very relevant since such shifts are very common in practice. The overall approach in the paper is of interest and the results in the paper provide a promising initial step towards novel measures of uncertainty that account for distributional shifts.


**Weaknesses:**

The interpretation of the s-value in (1) for scalar parameters is somehow clear since it measures the smallest divergence needed to change the sign of the parameter (assuming the parameter continuously depends on the distribution). However, the usefulness of its generalization to the multidimensional case in (11) needs further clarification. In the scalar case, the divergence needed to achieve a zero value reflects the divergence needed to have a change of sign. In the multidimensional case, it is not clear why measuring the divergence needed to achieve a zero vector valued \eta quantifies the instability.

The experimental results are not correctly described. It is hard to grasp the main takeaways of such results and the relationship with the theoretical results provided. In general, the paper presentation is rather poor and multiple results appear in the appendices. It is clear that the paper can be significantly improved.

The contribution of the proposed two-stage approach for transfer learning is hard to quantify. In general, the authors should compare their methods with existing techniques so that the paper's contributions can be better assessed.

**Questions:**

The definition in (2) is unclear in page 2. The authors should mention there that the set \mathcal{P} in (2) denotes joint distributions of Z and E, since in that page those distributions correspond to r.v. Z only.

In line 248 the authors mention "Our findings show that the average treatment effect is unstable with respect to changes in the marginal distribution of ‘age’, ‘education’, and ‘re75’. We also find that s-values conditional on ‘age’, ‘education’, ‘black’, ‘hispanic’, and ‘re75’ are non-zero, indicating that the average treatment effect can change its sign with a shift in the marginal distribution of these covariates (sX > 0.85)." Are those results shown in Figure 2?

The authors should improve the quality of figures that currently is rather poor.

**Limitations:**

The authors should improve the description of the limitations of the methods proposed as described in the "weaknesses" section.

---

> ### Author Rebuttal · Authors · 2023-08-05
>
> Thank you for the helpful suggestions and interest in our work. We hope that the additional evidence and our explanations will address your concerns.
>
> > The interpretation of the s-value in (1) for scalar parameters is somehow clear since it measures the smallest divergence needed to change the sign of the parameter (assuming the parameter continuously depends on the distribution). However, the usefulness of its generalization to the multidimensional case in (11) needs further clarification. In the scalar case, the divergence needed to achieve a zero value reflects the divergence needed to have a change of sign. In the multidimensional case, it is not clear why measuring the divergence needed to achieve a zero vector valued $\eta$ quantifies the instability.
>
> We agree that (11) does not have many direct applications. The s-value in (11) is just an intermediate result. In practice, the stability of scalar components of multidimensional parameters is of paramount importance (Section 5.1, Section 5.2, Section D.1). For example, in causal inference, we are often interested in the average treatment effect (a scalar component) in the presence of multiple nuisance parameters (confounders). Such an example is presented in Section 5.1.
>
> > The experimental results are not correctly described. It is hard to grasp the main takeaways of such results and the relationship with the theoretical results provided. In general, the paper presentation is rather poor and multiple results appear in the appendices. It is clear that the paper can be significantly improved.
>
> We are not sure what is meant by “not correctly described”. In the final version, we will include tables of s-values (see response to reviewer oQ9C) and improve the quality of the figures. We believe that this will improve readability. If this does not address your concern, we would appreciate additional input.
>
> Regarding the Appendix: We have decided to explain the conceptual ideas in the main paper and move more technical results to the appendix. The goal is to confer the main ideas and numerical evidence, without losing “tempo”. We believe that this is a stylistic choice and hope that the reviewer agrees with our reasoning.
>
> > The contribution of the proposed two-stage approach for transfer learning is hard to quantify. In general, the authors should compare their methods with existing techniques so that the paper's contributions can be better assessed.”
>
> Unfortunately, for the s-value itself, there exists no direct competitor (see rebuttal summary above).
>
> Based on your feedback we have evaluated a popular feature selection method, the Lasso. For the wine quality data, it selects all covariates while for the NSW data, it selects no covariates. The reason for this failure is that the Lasso selects based on feature importance in a prediction problem instead of stability in an estimation problem.
>
> The naive approach is to conduct semi-parametrically efficient transfer learning based on data on all covariates. This is an oracle procedure that provides an upper bound on other procedures. We show that our approach is competitive with this data-hungry oracle method while requiring much less data.
>
> > The definition in (2) is unclear in page 2. The authors should mention there that the set $\mathcal{P}$ in (2) denotes joint distributions of Z and E, since in that page those distributions correspond to r.v. Z only.
>
> We will address this and clarify that the set  $\mathcal{P}$ in (2) denotes joint distributions of Z and E in the final version.
>
> > In line 248 the authors mention "Our findings show that the average treatment effect is unstable with respect to changes in the marginal distribution of ‘age’, ‘education’, and ‘re75’. We also find that s-values conditional on ‘age’, ‘education’, ‘black’, ‘hispanic’, and ‘re75’ are non-zero, indicating that the average treatment effect can change its sign with a shift in the marginal distribution of these covariates ($s_X > 0.85$)." Are those results shown in Figure 2?
>
> Yes, the s-values can be obtained from Figure 2 by checking at which x-values the line cross zero, but we agree that it is more complicated than necessary. Following your feedback, we have decided to include a table with the s-values in the main paper (tables can be found in the response to reviewer oQ9C). Thank you for your input!
>
> > The authors should improve the quality of figures that currently is rather poor.
>
> We realize that some characters are only partially visible in Figures 2, 4 and 5. We will address this in the final version. We will also fix some color issues. If you have any other feedback, please let us know.
>
> > The authors should improve the description of the limitations of the methods proposed as described in the "weaknesses" section.
>
> We will expand the discussion of the weaknesses; in particular, we will describe more directly situations that might require other types of distributional stability values. For example, ongoing work covers shifts in $Y|X$. We believe there is room for a variety of new stability values that cover different types of distributional shifts. If you have additional feedback, please let us know.

---

> > ### Comment · Reviewer_buuF · 2023-08-15
> >
> > I would like to thank the authors for their responses. However, I still think the paper contribution is not significant enough for this conference. For instance, it seems the methods proposed are only useful for the scalar case.

---

> > > ### Author Response · Authors · 2023-08-15
> > > **Misunderstanding**
> > >
> > > Thank you for engaging with our review. We believe that there has been a misunderstanding.
> > >
> > > Our procedure can be helpful in the multi-parameter case. For example, ANOVA can be used to test hypotheses about multiple parameters. ANOVA requires being specific about the null hypothesis (for example that the group means in ANOVA are all equal to zero). For such a hypothesis, one can also compute s-values as in equation (11) with eta=0. For more details, please see Appendix Section D.
> > >
> > > Thus, the proposed method applies to and can be useful in multi-dimensional situations as well, but scientific applications are usually formulated as inferences on scalar parameters in the presence of (potentially high-dimensional) nuisance parameters, which is the main focus of the paper. The focus on scalar parameters in the presence of nuisance parameters is a stylistic decision that reflects scientific practice.
> > >
> > > To the best of our knowledge, we are the first to study the stability of parameters under various types of distributional shifts.

---

### Official Review · Reviewer_ktSh · 2023-07-05

**Soundness:** 4 excellent
**Presentation:** 3 good
**Contribution:** 2 fair
**Rating:** 3
**Confidence:** 4

**Summary:**

This work defines a novel statistical measure of the “stability” of a parameter in a distribution with respect to changes in that distribution. From a high level, it is defined as the minimum KL distance to a perturbation that flips the sign of the parameter. A method is given to calculate the s-value for mean parameters and several experiments are performed to show its utility.

**Strengths:**

The method extends to the directional case. I can see the application of this to average treatment effect to be useful in industry in online experimentation platforms. Strong arguments for the utility of s-value for doing robust transfer learning.

**Weaknesses:**

One weakness is that the calculation of the s-value uses a theorem that is only applicable to mean value parameters. The difficulty is that the definition uses an optimization problem over a set of functions rather than real-valued variables.

**Questions:**

Could you extend to more general cases by learning a flexible neural density approximator to P_0, and solving (3) by (constrained) variational inference?

**Limitations:**

Even though it’s addressed in the appendix, it seems non-practical to calculate the s-value for parameters other than the mean or functions of the mean.

The paper restricts discussion to distributions that are absolutely continuous with respect to the training distribution, although this is not much of a limitation as it covers most practical cases of interest.

---

> ### Author Rebuttal · Authors · 2023-08-05
>
> Thank you for your time in evaluating our manuscript. We hope that we can address your concerns with this rebuttal.
>
> > One weakness is that the calculation of the s-value uses a theorem that is only applicable to mean value parameters.
>
> Please note that we have many results that cover more general cases (e.g., Example 6 in the Appendix covers generalized linear models). We provide an extensive theory (Appendix D, E, and F) to cover a variety of real-world cases.
>
> > The difficulty is that the definition uses an optimization problem over a set of functions rather than real-valued variables.
>
> In statistical and machine learning problems, it is common to initially state a problem as an optimization over an infinite-dimensional space. For example, non-linear prediction can be stated as arg min_f E[(Y- f(X))^2]. In observational causal inference, under ignorability the parameter is a functional of an infinite-dimensional nuisance parameter. Thus, we believe that our approach is well within mainstream ML.
>
> From a practical side, our theoretical results allow us to re-cast the problem as finite-dimensional optimizations (sometimes even one-dimensional convex problems, see lines 127 and 146).
>
> > Could you extend to more general cases by learning a flexible neural density approximator to $P_0$, and solving (3) by (constrained) variational inference?
>
> In general, such a direct approach is possible and might lead to interesting new procedures. However, our proposed procedure has two advantages that may not be straightforward to implement with the reviewer’s approach. First, neural density estimation does not consider the particular functional form of the estimand. As Donsker-Varadhan shows, taking into account the functional form can reduce the infinite-dimensional problem to a finite-dimensional one. Secondly, density estimation usually has extremely slow convergence rates and (in our experience) might lead to unreliable estimates unless penalized very carefully.
>
> > Even though it’s addressed in the appendix, it seems non-practical to calculate the s-value for parameters other than the mean or functions of the mean.
>
> We respectfully disagree with this statement. From an optimization perspective, the question is not whether the parameter is a function of the mean; it is whether the parameter is a linear functional in the distribution space. Parameters of linear regression are non-convex functionals when viewed as functionals on the distribution space. Thus, the empirical example in Section 5.2 already captures this challenge. More general cases (such as empirical risk minimization) are treated in extensive detail in the appendix.

---

### Official Review · Reviewer_msmy · 2023-07-10

**Soundness:** 3 good
**Presentation:** 3 good
**Contribution:** 3 good
**Rating:** 6
**Confidence:** 4

**Summary:**

This paper proposes a measure to quantify the instability of a statistical parameter under distribution shifts, which calculates the minimal KL divergence to flip the sign of the estimated parameter. The authors demonstrate its usage in helping to collect target samples in transfer learning. The idea is clear and the theoretical analysis is solid, while the experiments seem a little inadequate.

**Strengths:**

1. The idea makes sense and the paper is easy to follow.
2. The proposed measure is novel, and the theoretical analysis is solid.
3. The proposed measure could help to collect data in transfer learning, and the two-stage method is simple but seems efficient.

**Weaknesses:**

Generally, I love this paper, but there are some drawbacks that stop me from giving a higher score:
1. The experiments seem a litter inadequate. I view this paper as technical work, but there are almost no baselines to compare in experiments. There are some recent papers sharing similar ideas (efficiently collecting data in transfer learning), and I think the authors should compare or at least mention them. For example:
* Data Shapley: Equitable Valuation of Data for Machine Learning. ICML 2019
* Shapley values for feature selection: The good, the bad, and the axioms
* Algorithms to estimate Shapley value feature attributions
* Towards Efficient Data Valuation Based on the Shapley Value
Also, maybe the authors could compare with some feature selection methods in experiments.

2. There are some typesetting problems in Figure 4 on page 9. Some contents are covered by figures.

3. The authors measure the instability via the effects on estimation. I wonder whether it is a better way to directly measure the model performance since model performance has already taken into consideration of the estimated parameters. For example, this paper: "Minimax Optimal Estimation of Stability Under Distribution Shift  Hongseok Namkoong, Yuanzhe Ma, Peter Glynn". I hope that the authors could discuss these two perspectives more.

**Questions:**

Please refer to weaknesses.

**Limitations:**

How could the proposed measure be used in neural networks or in large-scale problems?

---

> ### Author Rebuttal · Authors · 2023-08-05
>
> Thank you for your thoughtful feedback. We are excited to hear that you “love this paper”!
>
> > The experiments seem a litter inadequate. I view this paper as technical work, but there are almost no baselines to compare in experiments. There are some recent papers sharing similar ideas (efficiently collecting data in transfer learning), and I think the author should compare or at least mention them. For example: (...)
>
> Thank you for these references, they help us clarify differences. We will include a short discussion in the final version. Here is our view:
>
> For Shapley values, the goal is usually to evaluate the value of data (sets) or features in the context of prediction. In contrast, we are interested in parameter changes under distribution shift. As an example, if you want to predict whether someone has a headache you might want to take into account the situation an individual is in; but understanding whether the causal effect of taking an aspirin generalizes to a new situation is a different question.
>
> Regarding baselines: We are unaware of direct competitors to our approach. To the best of our knowledge, there exists no other method that measures the stability of a parameter with respect to various covariate shifts. Thus we compare it to an oracle procedure that gets to use additional data from the target distribution and conducts semi-parametrically efficient transfer using double-machine learning [1,2]. In our view, this is among the strongest oracles possible. Motivated by your question, we have tried feature selection based on the Lasso. As discussed in the rebuttal summary, feature selection based on the Lasso fails at our task of selecting the important variables for parameter transfer.
>
> > Also, maybe the authors could compare with some feature selection methods in experiments.
>
> Thank you for the suggestion. Based on your suggestion, we have applied a popular feature selection method (Lasso) to the experiment. For the NSW data set, Lasso selects no covariates and for the wine quality data set it selects all covariates. We attribute this failure to the fact that the Lasso quantifies feature importance in prediction problems, instead of sensitivity in parameter estimation problems.
>
> > There are some typesetting problems in Figure 4 on page 9. Some contents are covered by figures.
>
> Thank you. We will fix this in the final version.
>
> > The authors measure the instability via the effects on estimation. I wonder whether it is a better way to directly measure the model performance since model performance has already taken into consideration of the estimated parameters. For example, this paper: “Minimax Optimal Estimation of Stability Under Distribution Shift”
>
> Thank you for the thoughtful comment. We love the mentioned paper, since it employs a similar philosophy, albeit for prediction problems instead of parameter estimation problems. Let’s consider the example of fitting a model for causal inference. One parameter might correspond to the treatment, while the other parameters might correspond to nuisance parameters (such as the effect of age, education, socioeconomic status,...). Model performance would now measure the stability of the model with respect to shifts (including changes in age and education) and would be misleading if someone is just wondering whether the treatment would still work in a new situation.
>
> In short, evaluating model stability is useful for predictions under shifts, while parameter stability is useful for parameter estimation under shifts.
>
> ### References
> [1] Jin, Ying, and Dominik Rothenhäusler. "Tailored inference for finite populations: conditional validity and transfer across distributions." Biometrika (2023)
>
> [2] Chernozhukov, V., Chetverikov, D., Demirer, M., Duflo, E., Hansen, C., Newey, W., & Robins, J. (2018). Double/debiased machine learning for treatment and structural parameters.
>
> [3] Wager, S., & Athey, S. (2018). Estimation and inference of heterogeneous treatment effects using random forests. Journal of the American Statistical Association, 113(523), 1228-1242.

---

> > ### Comment · Reviewer_msmy · 2023-08-17
> >
> > Thank you for your detailed explanation. I updated my score to believe the following will be reflected in your revised manuscript (or camera-ready version).
> > * A discussion on relevant works measuring stability, e.g. “Minimax Optimal Estimation of Stability Under Distribution Shift”.
> > * A discussion on the choice of KL divergence, especially its potential shortcomings.

---

### Official Review · Reviewer_jftc · 2023-07-14

**Soundness:** 3 good
**Presentation:** 3 good
**Contribution:** 3 good
**Rating:** 7
**Confidence:** 4

**Summary:**

For a given data distribution P0 and family of distributions around it, mathcalP, the authors propose a measure of stability in estimating
a parameter theta(P) when there are distribution shifts within mathcalP.  The idea is that you have data P0, but might really be interested
in estimating theta for some P' != P0 that you might get when deploying your model. That is, from data P0, know how bad you will
be at estimating theta(P'). Sensitivity to changes depend on the parameter of interest, and the idea here is to quantify how much of a shift it
would take (within P, from P0) to change the sign of theta(P0). The resulting quantity s(theta, P0), which should maybe be caled s(theta, P0, mathcalP) is called the s-value, which takes values in [0,1] for 1 meaning "small shifts will change the sign of theta" to 0 meaning "the sign is stable".

The authors not only study marginal shifts, but also those resulting when one knows that some given conditionals will be fixed. Finally, they describe a two stage procedure for determining if it would be useful to collect test set samples for certain subsets of features (those for which s value is most unstable).

**Strengths:**

- distribution shift is discussed in ML for predictive tasks but less in statistical inference / parameter estimation settings. It's great that the authors are bringing together the two areas, and providing tools for reasoning about inferences in the real world where we do expect shifts
- Moreover, we don't always care just about absolute shifts, but instead qualitative things, like parameter sign changes, which the authors focus on
- The authors discuss the conditional case where shift is only limited to some features, which helps us study more specific instances of the problem as well as gain statistical efficiency when we do have some knowledge

**Weaknesses:**

The weaknesses are not in the mathematical method nor experiments, but more so in the discussion and contextualization of this work.

- I think the iterative two stage procedure for selecting features to collect test data for is a pretty cool use case, but there is some lacking discussion about when this would actually be feasible in the real world. Please further discuss when one should or should not be able to do this.

- The authors somewhat quickly dismiss influence-function (IF) based estimation in the Related Work, but my strong feeling is that a more thorough discussion of the relationship to this field is necessary. IF based estimation is not just about robustness to outliers, but actually has a fairly deep connection to this work where influence functions show up in the functional derivative of a parameter to be estimated with respect to an underlying distribution. Moreover, just like the authors of this work consider the case of parameter sensitivity conditionally on certain factors of the joint distribution, the IF literature likewise considers projections that describe parameter sensitivity to changes in distributions when some factors are fixed. I admit that the IF literature is somewhat dense, but I believe it would be useful to state some relationships even at high level to the derivatives and Von-Mises ("distributional taylor") expansions in e.g. Kennedy's review here: https://arxiv.org/pdf/2203.06469.pdf

---- just some writing comments below ---

- I assume figure 2 is about the data NSW given the legend mentioning demographics, but the figure did not explicitly mention this. The figure caption should mention the name of the dataset to avoid confusion.

- Figure 3 does not mention which data it is about in the caption, and one has to work backwards to the NSW experiment to find out which data the figure is about by searching for the text "Figure 3". Please mention the data in the figure caption.

- when you say "We employ Jin and Rothenhäusler [22]’s transfer procedure to estimate", it would be helpful to be able to read what the method is from a re-statement in your paper rather than having to open the citation, since it's a detail that is part of your experiments.

- in lines 59-61, "We discuss how our ... same to re-estimate the parameter under shifted distribution", it would be correct to change "under shifted distribution" to either "under a shifted distribution" or "under shifted distributions".

- The remark in 63-70 could be moved to the end of work since it sort of interrupts the flow

**Questions:**


- The two-stage procedure involves testing the stability of various subsets of features and seeing if there are some for which it would help to collect samples from the test set. Since it would be usually not possible to do this kind of search in the real world, what would be the practical take-away for how to make use of this in non synthetic studies? Did I understand correctly that testing a given subset Xs requires collecting it from the test set? It's fine to study this ideal case in a research paper, but it seems like it necessitates more discussion.

- NSW experiment: your initial estimator has a std dev of 492 which is large for an effect size of 820. How was the std dev estimated? Is this the expected ATE std dev using this method on this data?

- There is a claim in the paper that "full transfer" doesn't provide much gain over "partial transfer". This probably needs some more explanation. Is the claim that  using Pproj to estimate theta is as good as using Ptest? This must only be true under some assumptions. Even if these are mentioned implicitly when describing the data, it would be helpful for the authors to restate what might cause this phenomenon. It's fine to report this phenomenon if it does hold on your data, but please contextualize.

**Limitations:**

Yes.

---

> ### Author Rebuttal · Authors · 2023-08-05
>
> We appreciate your careful and thorough review. We are glad that you find it “great that the authors are bringing together the two areas” [distribution shift in ML for predictive tasks and estimation problems in statistics]!
>
> Thank you for your thoughtful comments on writing, we will address them in the final version of the manuscript.
>
> > I think the iterative two-stage procedure for selecting features to collect test data is a pretty cool use case, but there is some lacking discussion about when this would actually be feasible in the real world. Please further discuss when one should or should not be able to do this.
>
> In the following, we will discuss an example from the social sciences since applied researchers from political science have signaled interest in our procedures.
>
> Researchers are often interested in a causal effect estimate for a new location. For some covariates such as age and education, partial data is available via surveys such as American National Election Study (ANES) or Cooperative Election Study (CES). Additional partial data can be cheaply obtained via Amazon Mechanical Turk. However, some covariates are hard to collect, since they require running a study in the new location. Our numerical results show that the proposed approach can help prioritize data collection. This may drastically reduce the cost compared to running full-scale replication studies.
>
> We will add this discussion to the final manuscript.
>
> > The authors somewhat quickly dismiss influence-function (IF) based estimation in the Related Work, but my strong feeling is that a more thorough discussion of the relationship to this field is necessary. IF-based estimation is not just about robustness to outliers, but actually has a fairly deep connection to this work where influence functions show up in the functional derivative of a parameter to be estimated with respect to an underlying distribution.
>
> We agree that functional derivatives are closely related to our work and apologize if the “related work” section gave this impression. In fact, the theory in the Appendix is based on functional derivatives of the parameter. We will rephrase the corresponding sentences in the final version.
>
> > The two-stage procedure involves testing the stability of various subsets of features and seeing if there are some for which it would help to collect samples from the test set. Since it would be usually not possible to do this kind of search in the real world, what would be the practical take-away for how to make use of this in non synthetic studies? Did I understand correctly that testing a given subset Xs require collecting it from the test set? It’s fine to study this ideal case in a research paper, but it seems like it necessitates more discussion.
>
> To compute the s-value it is *not* necessary to have data from the test set. Searching over several subsets can be done without having additional data. Only for conducting the transfer procedure, one needs partial data from the test set. For a more concrete example discussing a use-case, see above.
>
> > NSW experiment: your initial estimator has a std dev of 492 which is large for an effect size of 820. How was the std dev estimated? Is this the expected ATE std dev using this method on this data?
>
> We chose the NSW data set since it is the most widely used data set in observational causal inference. Standard deviations for classical procedures can be found in [1], and range between 500 and 1100. Our baseline procedure is based on SOTA cross-fitted doubly-robust machine learning and thus performs a bit better than classical approaches.
>
> We estimated the standard deviation by the empirical standard deviation of the cross-fitted influence function, which is the standard implementation in the R-package GRF [2].
>
> > There is a claim in the paper that “full transfer” doesn’t provide much gain over “partial transfer”. This probably needs more explanation. Is the claim that using Pproj to estimate theta is as good as using Ptest? This must only be true under some assumptions. Even if these are mentioned implicitly when describing the data, it would be helpful for the authors to restate what might cause this phenomenon. It’s fine to report this phenomenon if it does hold on your data, but please contextualize.
>
> Thanks for helping us clarify this. Our intuition is that in these cases most of the shift is in observables covariates $X$ (while $Y|X$ is roughly invariant). The s-value captures which types of $X$-shifts the parameter is sensitive to. Full transfer does not improve performance, since it additionally updates the distribution along directions that the parameter is not sensitive to.
>
> ### References
> [1] Dehejia, R. H., & Wahba, S. (1999). Causal effects in nonexperimental studies: Reevaluating the evaluation of training programs. Journal of the American Statistical Association, 94(448), 1053-1062.
>
> [2] Athey, Susan, Julie Tibshirani, and Stefan Wager. "Generalized random forests." (2019): 1148-1178.

---

### Official Review · Reviewer_oQ9C · 2023-07-19

**Soundness:** 3 good
**Presentation:** 2 fair
**Contribution:** 3 good
**Rating:** 6
**Confidence:** 3

**Summary:**

This work introduces the s-value, a novel metric to measure the stability of statistical parameters. It is defined as the exponential of minus the largest KL-divergence for which the statistical parameter is 0. The smaller the s-value the more stable the parameter is. When the statistical parameter is the mean of a random variable, the s-value is shown to have a simple expression. The paper also introduces the less conservative directional s-value, which constrains the type of distribution shift possible.

**Strengths:**

Many methods exist to find the worst case distribution shift within some radius as a way to obtain a robust estimator. This paper instead is finding the largest radius up to which a parameter is robust, and then use this value to derive a measure of stability. I find this approach novel and I can imagine this paper having some impact in its area.

The idea behind the paper and the derivations are sound, I also appreciate the exhaustive appendix.

**Weaknesses:**

I found the experiments hard to interpret, e.g. in figure 4 and 5, what is $\beta$? Maybe you could improve the legend for those figures to make them easier to interpret. In figure 4, why are the scales so different ([-2,2] vs [-200,100]). It might have been more valuable to dive into a single example and describe it more in depth. Considering the main focus of this paper, I would also suggest showing the s-values for different parameters e.g. in a table.


Given the venue, I was expecting that the estimation of the s-values for parameters defined via risk minimization would be in the main paper.


Overall, I appreciate the quality of this paper, but I would have liked to see more clearly how impactful the s-values can be in concrete cases. Achieving this might simply require improving the experiment section to address it to a broader audience.

**Questions:**

How useful would s-values be for non-convex models? In the appendix, it is mentioned that a small s-value cannot be necessarily interpreted as a proof of stability in that case. What results would you expect to obtain if you were to run your parameter transfer experiment using a non-linear model?

**Limitations:**

I think limitations could be better addressed by e.g. adding a paragraph in the appendix.

---

> ### Author Rebuttal · Authors · 2023-08-05
>
> Thank you for your thorough review. Also, we are glad to hear that you “appreciate the quality of this paper”.
>
>
>
> > I found the experiments hard to interpret, e.g. in Figure 4 and 5, what is beta? Maybe you could improve the legend for those figures to make them easier to interpret. In Figure 4, why are the scales so different. It might have been more valuable to dive into a single example and describe it in more depth. Considering the main focus on the paper, I would also suggest showing the s-values for different parameters, e.g. in a table.
>
> Thank you for the thoughtful feedback. We will improve the descriptions for the final version. In Figure 4, the scales are very different since the plots correspond to different covariates. Following your suggestion, we will show the s-values in a table.
>
> ### S-values for NSW data set
>
> |Feature| Age  | Education|  Black |   Hispanic | Married |Nodegree|Re75 |
> | -----------| ----------- | ----------- |----------- | ----------- | ----------- |----------- |----------- |
> |Directional s-value|  0.97 | 0.91|0.52 | 0.54|0 |0 |0.96 |
>
>
> ### S-values for wine quality data set (parameter "pH")
>
> | Feature | fixed.acidity | volatile.acidity | citric.acid | residual.sugar | chlorides | free.sulfur.dioxide | total.sulfur.dioxide | density | pH  | sulphates | alcohol |
> | ------- | ------------- | ---------------- | ----------- | -------------- | --------- | ------------------- | ------------------- | ------- | --- | --------- | ------- |
> | Directional s-value | 0.86          | 0.81             | 0.65        | 0.83           | 0.94      | 0.55                | 0.8                 | 0.83    | 0.97 | 0.97      | 0.88    |
>
> ### S-values for wine quality data set (parameter "density")
>
> | Feature              | fixed.acidity | volatile.acidity | citric.acid | residual.sugar | chlorides | free.sulfur.dioxide | total.sulfur.dioxide | density | pH   | sulphates | alcohol |
> | -------------------- | ------------- | ---------------- | ----------- | -------------- | --------- | ------------------- | ------------------- | ------- | ---- | --------- | ------- |
> |Directional  s-value              | 0.80          | 0.94             | 0.83        | 0.81           | 0.78      | 0.81                | 0.93                | 0.84    | 0.79 | 0.9       | 0.98    |
>
> > How useful would s-values be for non-convex models? In the appendix, it is mentioned that a small s-value cannot be necessarily interpreted as a proof of stability in that case. What results would you expect to obtain if you were to run your parameter transfer experiment using a non-linear model?
>
> To summarize, we believe that the s-value is useful for cases that are most important for empirical applications, despite computational difficulties for non-convex models. Let’s discuss this in more detail.
>
> We expect s-values to be accurate for small distributional changes since in that case the distribution change is well-approximated by a Taylor expansion. For large distributional changes, the s-value may exhibit high variance and may not be interpreted as proof of stability. However, in this case, the weights in transfer learning procedures will get extreme, leading to large confidence intervals. From a practical perspective, transfer learning in such settings is challenging if not impossible.
>
> > Given the venue, I was expecting that the estimation of s-values for parameters defined via risk minimization would be in the main paper.
>
> From a conceptual perspective, there is no new step from what Section 3 covers in the main paper. Our goal with the main paper is to lay out the ideas and concepts as clearly as possible, without losing “tempo”. We believe that this is a question of style, and hope that the reviewer appreciates our reasoning.
>
> > Overall, I appreciate the quality of this paper, but I would have liked to see more clearly how impactful the s-values can be in concrete cases. Achieving this might simply require improving the experiment section to address it to a broader audience.
>
> As written above, we will update the description of the figures in the final version.
>
> Regarding “impact”: The NSW “LaLonde” data set [1] is one of the most widely used data sets in observational causal inference over the last few decades. On this data set, our two-stage procedure performs as well as a “potentially unobtainable” oracle procedure and improves more than 50% over a naive baseline. Thus, we believe that a large audience will be able to appreciate the results of our analysis. However, we would be happy to receive further input and address them in the final version.
>
> ### References
> [1] Dehejia, R. H., & Wahba, S. (1999). Causal effects in nonexperimental studies: Reevaluating the evaluation of training programs. Journal of the American Statistical Association, 94(448), 1053-1062.

---

> > ### Comment · Reviewer_oQ9C · 2023-08-21
> >
> > I thank you for your clarifications. I decide to keep my score but raise my confidence.

---

### Official Review · Reviewer_qNys · 2023-07-27

**Soundness:** 3 good
**Presentation:** 3 good
**Contribution:** 3 good
**Rating:** 7
**Confidence:** 3

**Summary:**

This paper proposes a novel metric, called the s-value, for evaluating the stability of statistical parameters with respect to distributional shifts. This metric is based on a variational problem involving the KL divergence between the target distribution and the shifted distribution, which can be solved via an equivalent one-dimensional convex problem. The authors also introduce the notion of directional s-value that quantifies the instability of directional shifts. Moreover, consistency and asymptotic normality results are proven for the plug-in estimators of these s-values. Finally, the authors illustrate the interest of the s-values on some real datasets.

**Strengths:**

1. I found the proposal novel and interesting.
2. Quantifying the stability of statistical findings under distributional shifts is an important question and this work takes a first step towards answering this question.
3. The paper is well-organized and easy to follow.

**Weaknesses:**

1. The treatment for the consistency of $\hat s_E(\mu, P_n)$ seems to be weak. In particular,
  (1) Estimating the conditional expectation $E[Z | E]$ is a challenging problem, especially when the dimension of $E$ is large as in Example 4. The uniform convergence assumption (Assumption 1 in Appendix C) seems to be too strong.
  (2) The rate of convergence of $\hat f_n(E)$ can be very slow when the dimension is large. It is of interest to know how the rate of convergence of $\hat s_E(\mu, P_n)$ depends on the one of $\hat f_n(E)$.
2. The practical usefulness of this metric is questionable. In the experiments the authors only computed a few s-values without giving empirical evidence supporting the reasonableness of these numbers. For example, is a statistical parameter with a large s-value really more sensitive to distribution shifts than one with a small s-value? It would be good to perform at least simulation studies to investigate this question.

**Questions:**

1. Can you prove the consistency of $\hat s_E(\mu, P_n)$ under a weaker assumption on $\hat f_n(E)$?
2. How does the rate of convergence of $\hat s_E(\mu, P_n)$ depends on the one of $\hat f_n(E)$?
3. Can you provide empirical results supporting the practical usefulness of s-values? For example, is a statistical parameter with a large s-value really more sensitive to distribution shifts than one with a small s-value?

**Limitations:**

Yes

---

> ### Author Rebuttal · Authors · 2023-08-05
>
> Thank you for your thoughtful remarks.
>
> > Can you prove the consistency of $\hat s_E (\mu, P_n) $ under a weaker assumption on $\hat f_n(E)$?
>
> Yes, such a result can be obtained under $L_p$ convergence and a boundedness assumption, with small modifications to the current proof. We are happy to relax this in the final version of the manuscript if desired.
>
> > How does the rate of convergence of $\hat s_E(\mu,P_n)$ depends on the one of $\hat f_n(E)$?
>
> If $\hat f_n$ has slow convergence (slower than $1/\sqrt{n}$), then the naive estimation of $\hat s_E$ also has a slow convergence rate. This can be improved, as explained below.
>
> In Lemma C.4 in the Appendix, we derive a debiasing procedure that shows that even if the convergence rate of $\hat f_n$ is slow (e.g. $n^{-1/4}$), one can still obtain a fast $1/\sqrt{n}$ rate of convergence for $\hat s_E$. The proof is based on de-biasing techniques from the semiparametric literature that have recently attracted considerable attention under the name of “double machine learning” [1].
>
> > Can you provide empirical results supporting the practical usefulness of s-values? For example, is a statistical parameter with a large s-value really more sensitive to distribution shift than one with a small value?
>
> Checking whether s-values are “reasonable” is best done with a small toy example where it is clear what the correct answer should be. In the Appendix, we give such a toy example (Figure 6 and Table 1) based on the famous “Anscombe quartet”. We realize that it might be helpful to move this example to the main paper for the final version.
>
> Regarding practical usefulness, the applied community has signaled interest in the proposed procedures. For example, Egami and Devaux [2] estimate the KL divergence between multiple surveys (such as ANES or CES) and use this to derive appropriate thresholds for s-values. Other ongoing applied work focuses on shifts in $Y|X$ “concept drift”.
>
> ### References
> [1] Chernozhukov, V., Chetverikov, D., Demirer, M., Duflo, E., Hansen, C., Newey, W., & Robins, J. (2018). Double/debiased machine learning for treatment and structural parameters.
>
> [2] Devaux, M., & Egami, N. (2022). Quantifying robustness to external validity bias. Available at SSRN 4213753.

---

> > ### Comment · Reviewer_qNys · 2023-08-17
> > **Response to Rebuttal**
> >
> > Thank you for your responses. I have no more questions and will maintain my rating.

---

### Official Review · Reviewer_JAUw · 2023-07-28

**Soundness:** 3 good
**Presentation:** 4 excellent
**Contribution:** 2 fair
**Rating:** 5
**Confidence:** 3

**Summary:**

This paper proposes method to quantify instability of a statistical parameter with respect to pertubrations around the KL divergence ball. This has implications to detect where statistical conclusions no longer hold when there is a distribution shift. The authors show this metric across both overall and directional shifted. The paper provides a two-step transfer learning strategy over two datasets to demonstrate the effectiveness of using stability as a measure of where to collect extra data for transfer learning.

**Strengths:**

Tthe paper was well written, with its objectives and motivations clear.  Theorems and mathematical notation were easy to follow. The examples in Section 3.2 illustrated well how this metric is concretely used for different distributions. It is evident that  s-values can be useful in determining when to re-train models or re-estimate statistical queries, depending on the stability of a particular parameter. Detecting under which variables shifts occurs is an important open problem that the authors provide clear insight in, as well as a procedure to use s-values for improved transfer learning.

**Weaknesses:**

However, while an interesting and intuitive idea, it does not seem that it provides significant improvements over a transfer learning approach with all covariates. In Figures 3 and 5, transfer learning outperforms the naive approach, but this is an expected result. What is the reason for a partial transfer if a full transfer works better than or just as well? I would also be interested to see an experiment where a greater amount of data can be collected to improve the transfer learning (aka a higher $\alpha$ value for the second experiment).

It would be also helpful to provide recommendations for practitioners for what it means when a parameter is stable (is there a general threshold of $s$ at which transfer learning works well over a parameter works well?) Is the $s_X > 0.85$ threshold as shown in the paper recommended?


**Questions:**

There are some uncertainties about the experimental methodology. Why were these particular datasets chosen? Something with an average treatment effect seems like it would be useful to be seen in medical datasets for a particular intervention? Would be interested to see if there would be significant improvements over standard transfer learning with all covariates if this procedure was done with datasets with more unstable covariate variables. It would be helpful to see a broader set of experiments, or a more thorough analysis of which covariates are helpful in this case or not.

**Limitations:**

Yes

---

> ### Author Rebuttal · Authors · 2023-08-05
>
> Thank you for the thoughtful feedback and comments.
>
> > What is the reason for a partial transfer if a full transfer works better than or just as well?
>
> Cost! Full transfer relies on having data on all covariates. Some of the data might be very expensive to collect. Partial data is often available in the form of surveys (e.g. ANES or CES) or can be collected via Amazon Mechanical Turk [1]. Our numerical results show that s-values help decide which covariates one should obtain for transfer learning. In our view, these are exciting results that may help reduce the cost of planning and running replication studies.
>
> > I would also be interested to see an experiment where a greater amount of data can be collected to improve the transfer learning (aka a higher alpha value for the second experiment)
>
> Sure! We have updated the figures and included them in the pdf attached to "Rebuttal Summary". Our conclusions remain unchanged.
>
> > It would also be helpful to provide recommendations for practitioners for what it means when a parameter is stable (is there a general threshold of s at which transfer learning works well over a parameter works well?) Is the $s_X > .85$ threshold as shown in the paper recommended?
>
> For the data sets we have worked with, the threshold .85 has worked well. However, in the end, this is an empirical question and depends on the subject of study. There is some exciting parallel work by Egami and Devaux [1] who build a library of reference values based on survey data sets. Their empirically validated thresholds are similar to ours.
>
> > Why were these particular datasets chosen?
>
> The NSW data set (often referred to as the “LaLonde data”) is the most commonly studied data set in causal inference. The wine quality data set was chosen since it has appeared in recent work studying distribution shifts [2]. We have chosen these data sets to ensure high familiarity of the target audience with the data sets.
>
> > Would be interested to see if there would be significant improvements over standard transfer learning with all covariates if this procedure was done with datasets with more unstable covariate variables. It would be helpful to see a broader set of experiments or a more thorough analysis of which covariates are helpful in this case or not.
>
> Thank you for the suggestion. We suspect this might be the case since our procedure ignores small shifts in unimportant variables, stabilizing the transfer learning procedure. We consider this an exciting direction for future research. For the current paper, we focus on the foundations of the method, which include theoretical guarantees and showing that it can perform as well as an oracle procedure in practice.
>
> ### References
> [1] Devaux, M., & Egami, N. (2022). Quantifying robustness to external validity bias. Available at SSRN 4213753.
>
> [2] Podkopaev, A., & Ramdas, A. (2021). Distribution-free uncertainty quantification for classification under label shift. In Uncertainty in Artificial Intelligence (pp. 844-853). PMLR.

---

### Author Rebuttal · Authors · 2023-08-05

# Rebuttal Summary

We thank you for your constructive feedback. We appreciate that you find the approach “novel and interesting” (qNys), that you “appreciate the quality of this paper” (oQ9C), that the “theoretical analysis is solid” (msmy), that it is “a promising initial step towards novel measures of uncertainty” (buuF) and that you see “strong arguments for the utility of s-value for doing robust transfer learning” (ktSh).

In the following, we will address two points that the majority of the reviewers have brought up as weaknesses: [1] practicality and feasibility, and [2] empirical evaluation.

## Practicality and feasibility

Several reviewers have asked for better contextualization, in particular, whether partial test data is available in practice, and how “impactful” stability values can be in practice.

Recall that the proposed workflow is to (i) evaluate the instability of the parameter with respect to shifts in different covariates; and (ii) update the parameter using data from the covariates flagged in the first step. Our procedure focuses on the first step. The second step can be conducted using existing procedures. In the following, we will discuss an example from the social sciences since applied researchers from political science have signaled interest in our procedures.

Researchers are often interested in a causal effect estimate for a new location. For some covariates such as age and education, partial data is available via surveys such as American National Election Study (ANES) or Cooperative Election Study (CES). Additional partial data can be cheaply obtained via Amazon Mechanical Turk. However, some covariates are hard to collect, since they require running a study in the new location. Our numerical results show that the proposed approach can help prioritize data collection. This may drastically reduce the cost compared to running full-scale replication studies.

As a side note, there is exciting empirical work by Egami and Devaux [1] who build a library of reference stability values based on survey data sets. They recommend thresholds for s-values based on empirical investigations of how data sets change between settings.


## Empirical evaluation

Overall, the reviewers have (i) asked about additional baselines, (ii) asked for a larger range of alpha in plots, and (iii) identified typesetting problems in some of the figures.

(i) Baselines

To the best of our knowledge, there exists no other method that measures the stability of a parameter with respect to various covariate shifts. Thus, there is no direct competing method. This is why we have compared our procedures to an oracle procedure that has access to additional data.

As suggested by the reviewers, we have evaluated a baseline based on feature importance. For the wine quality data set, Lasso selects all features, and for the NSW data set, Lasso selects no covariates. Thus, for NSW the performance of Lasso is equal to “Training”, while for “wine quality”, the performance of Lasso is equal to “Full Transfer”.

Why does Lasso feature selection fail? It solves a different problem; it captures the *feature importance in a prediction problem*, whereas our stability value captures the *sensitivity in a parameter estimation problem*.

(ii) A larger range of alpha

In the pdf attached to this summary, you find a plot corresponding to a larger range of alphas. Our conclusions remain unchanged.

(iii) Typesetting problems in some figures

The reviewers have criticized that some characters were only partially visible in Figures 2, 4 and 5. We will fix this in the final version of the manuscript. In addition, we will fix some color issues.


## References

[1] Devaux, M., & Egami, N. (2022). Quantifying robustness to external validity bias. Available at SSRN 4213753.

---

### Comment · Area_Chair_FmME · 2023-08-19
**Author's response to your concerns**

Dear reviewers,

The authors have responded to your questions and reviews. Since the discussion is coming to an end, we are wondering if you could kindly take a look and see whether the responses address your concerns and whether you'd like to update/maintain your initial rating.

Best,
AC

---

### Decision · Program_Chairs · 2023-09-21

**Decision:**

Accept (poster)

**Comment:**

This paper proposes a novel metric, called the s-value, for evaluating the instability of statistical parameters with respect to distributional shifts. The paper contains many interesting and novel ideas and is well-written. There are some concerns regarding the computational challenges in evaluating this metric as well as as the quality of numerical experiments and the authors are encouraged to take into account the related suggestions by the reviewers in the revision of their paper.